# Structural and immunological characterization of the H3 influenza hemagglutinin during antigenic drift

Rebeca de Paiva Froes Rocha[1,2,8], Ilhan Tomris [3], Charles A. Bowman[1], Emma Stevens [4,5], Jason Kantorow [4,5], Corinna M. Plitt[6], Weiwei Peng[6], Svearike Oeverdieck [1], Thales Galdino Andrade[1], James A. Ferguson[1], Diana D. Jung[1], Rafael Elias Marques[2], Sander Herfst [7], Joost Snijder [6], Sirupa Chakraborty [4,5], Alba Torrents de la Peña[1], Zachary T. Berndsen [1,9] ✉, Robert P. de Vries [3] ✉ & Andrew B. Ward [1] ✉

The quest for a universal influenza vaccine holds great promise for mitigating the global burden of influenza-related morbidity and mortality. However, challenges persist in identifying conserved epitopes capable of eliciting robust and durable immune responses. In this study, we explore the influence of glycan evolution on H3 hemagglutinin from 1968 to present day and its impacts on protein structure, antigenicity and immunogenicity by using computational, biochemical and biophysical techniques. Structural characterization of HK/68 and Sing/16 by cryo-electron microscopy shows that while HK/68 is resistant to enzymatic deglycosylation, removal of glycans destabilizes the hyperglycosylated head and membrane-proximal region in Sing/16. Furthermore, the appearance of glycans in Sing/16 hemagglutinin head domain shifts the polyclonal immune response upon vaccination to target the esterase and stem. These insights expand our understanding of glycans beyond their role in protein folding and highlight the interplay among glycan integration and immune recognition to design a universal influenza vaccine.

Hemagglutinin (HA), the most abundant glycoprotein on the surface of influenza viruses, serves as the primary antigenic target for neutralizing antibodies[1]. Consequently, genetic mutations in HA can drive antigenic drift, undermining pre-existing immunity and leading to vaccine mismatches[2]. In addition to genetic mutations, changes in the asparagine (N)-linked glycosylation pattern of HA contribute to immune evasion and have been shown to play a critical role in the emergence and spread of influenza A viruses (IAV) in humans. Notable examples include the antigenicity drop of the H3N2 strain in 2003 and the emergence of the 2009 H1N1 swine flu pandemic[3,4].

[1]Department of Integrative Structural and Computational Biology, The Scripps Research Institute, La Jolla, CA, USA. [2]Brazilian Biosciences National Laboratory (LNBio), Brazilian Center for Research in Energy and Materials (CNPEM), Campinas, Brazil. [3]Department of Chemical Biology and Drug Discovery, Utrecht Institute for Pharmaceutical Sciences, Utrecht University, Utrecht, The Netherlands. [4]Department of Chemical Engineering, Northeastern University, Boston, MA, USA. [5]Department of Chemistry and Chemical Biology, Northeastern University, Boston, MA, USA. [6]Biomolecular Mass Spectrometry and Proteomics, Bijvoet Center for Biomolecular Research, Utrecht Institute of Pharmaceutical Sciences, Utrecht University, Utrecht, The Netherlands. [7]Department of Viroscience, Erasmus Medical Center, Rotterdam, The Netherlands. [8]Present address: Department of Molecular and Cell Biology, University of California, Berkeley, CA, USA. [9]Present address: Department of Biochemistry, the University of Missouri, Columbia, MO, USA. ✉e-mail: zberndsen@missouri.edu; r.vries@uu.nl; andrew@scripps.edu

N-linked glycosylation is a critical co-translational and post-translational modification that impacts the structure and immunogenicity of viral proteins, including influenza HA[5]. For instance, adding specific glycans during viral replication is essential for the proper folding and trafficking of HA[6]. Glycans also shield the protein from the host immune system by preventing antibody recognition and promoting viral infectivity and replication. Thus, the accumulation of glycans in the receptor binding site (RBS) or RBS-proximal region is a strategy that the virus uses for immune evasion and to tune sialic acid receptor recognition by different glycoforms[7]. At the same time, changes in the number and location of glycans can also expose new epitopes that can be targeted by the immune system[8,9]. Previous studies have shown that the absence of a recently acquired glycan in a well-matched H3N2 vaccine resulted in ferrets and humans being unable to produce antibodies that efficiently neutralized circulating glycosylated viruses[3]. Thus, the interplay between HA glycosylation and the host immune response is critical to IAV pathogenesis and strain emergence.

Influenza HA exhibits a varying number of PNGS in both the variable head and the conserved stem region, dependent upon the specific virus strain and subtype. A PNGS is defined by the presence of the canonical Asn-X-Ser/Thr sequon, where X is any residue except proline. Analysis of the peptide sequence of hemagglutinins has shown that H3 circulating human strains have doubled the number of PNGS since its zoonotic emergence in 1968, with a particular concentration around the RBS[9–12]. Importantly, the presence of a PNGS is not sufficient to guarantee glycan occupancy[13]. While sequence analysis can be useful in predicting glycan occupancy on the HA surface, mass spectrometry-based approaches are necessary to accurately quantify and characterize the glycoforms present[14,15]. Although these methods can provide valuable insights, they do not offer information on the structural changes that occur as the glycosylation on HA evolves. Furthermore, even though crystallographic studies have provided some insights into how glycans affect epitope accessibility and recognition by antibodies, these studies have not fully elucidated how glycans dynamically impact HA protein stability[3,16].

Here, we used cryo-EM, polyclonal epitope mapping, computational modeling, and biochemical approaches, including glycoproteomics and immunoassays, to investigate the structural and functional consequences of glycan incorporation on H3 HA over the past five decades. Our analysis revealed that incorporation of PNGS, particularly in the globular head domain, shifts glycoform composition toward a higher mannose content and affects the stability of HA. Cryo-EM reconstructions demonstrated that enzymatic deglycosylation resulted in significant destabilization of the more contemporary H3 HA, Sing/16, whereas the 1968 H3 HA remained structurally intact following glycan removal. This difference is most pronounced in the hyper-glycosylated head domain, where glycan accumulation not only affects structural integrity but also has a substantial impact on epitope exposure and accessibility. These findings underscore the structural, functional, and stabilization impact of glycan addition on influenza HA and have direct implications for rational vaccine design and manufacturing processes.

## Results

### Emerging PNGS and substitutions in H3 viruses over the years
HA is a trimeric glycoprotein with each monomer composed of a globular head containing a vestigial esterase subdomain in its lower portion, and a stem domain[17] (Fig. 1A). Mutations that introduce N-linked glycosylation sites in the HA glycoprotein play a critical role in protein folding and immune evasion for human H3N2 viruses. This process may result in the addition of N-linked glycans, which can lead to substantial antigenic changes. Notably, the addition of glycans in H3 HAs is estimated to occur every 5 to 7 years and has been shown to influence vaccine efficacy[10]. To investigate the trends in the accumulation of glycans on HA over the years, we analyzed the presence of PNGS within circulating strains spanning 1968–2022. We analyzed N-glycosidic linkages to the Asn residue of the glycosylation motif Asn-X-Ser/Thr accumulated on HA based on the analysis of ~ 11,000 sequences of naturally occurring H3 strains using the code previously described by Wu et al.[3]. Specifically, we aligned the sequences the of ~ 11,000 naturally occurring HA H3 strains and analyzed N-glycosidic linkages to the Asn residue of the glycosylation motif Asn-X-Ser/Thr accumulated through time.[3]

Figure 1B illustrates the calculated frequencies of PNGS of the total number of circulating strains each year. Raw values and percentages can be found in Supplementary Table 1. From 1968 –2022, we observe the number of PNGS per protomer on HA went from 7 in 1968 (Fig. 1B, C) to 12 in 2022 (Fig. 1B, C). PNGS such as N6 and N7, found on HA's stem region, occur at a very low frequency, and therefore were not always represented (Supplementary Table S1). PNGS such as N8, N22, N38, N165, N285 and N483 have been present on HA since 1968 and persist until 2022 (Fig. 1B and Supplementary Fig. S1). Interestingly, these PNGS all appear on the vestigial esterase or stem region of HA except for N165, which is in the globular head at the interface between protomers (Fig. 1C). Aside from the 5 conserved PNGS appearing on HA's vestigial esterase or stem region (Fig. 1B, C and Supplementary Fig. S1), the number of additional PNGS in these regions has remained limited, never exceeding 2 per year, except in 1990–1991, when N45 briefly appeared in 20% of circulating strains alongside N81 and N63 (Fig. 1B and Supplementary Fig. S1).

By analyzing the glycans present on A/Hong Kong/1/1968 (HK/68) and comparing them to A/Singapore/INFIMH/16 (Sing/16), we observe that PNGS on HA's globular head, such as N126, N133 and N246, appear in 1974, 1996, and 1982, respectively, and persist until 2022 (Fig. 1C and Supplementary Fig. S1). On the other hand, the appearance of PNGS such as N122 and N144 (Fig. 1B, C), tend to fluctuate throughout the years (Fig. 1C and Supplementary Fig. S1). Interestingly, N158, a key molecular determinant of antigenic distancing between influenza strains and a key residue impacting HA sialic acid binding specificity[18–20], is found in < 40% of the circulating strains from 2014 to 2020 and completely disappears in the following years. This is in accordance with previous studies suggesting that H3N2 circulating strains after 2021, are expected to eliminate a glycosylation site at N158, impacting viral cell entry and influencing escape from vaccine-elicited antibodies[21].

Collectively, our data suggest PNGS appearing on the vestigial esterase and stem region tend to emerge less frequently and be more persistent when compared to PNGS appearing on the globular head (Fig. 1B, C and Supplementary Fig. S1). By analyzing the spatial distribution of PNGS in relation to residue conservation, we observed that PNGS predominantly emerge in non-conserved regions, specifically within the head domain (Fig. 1D).

In addition, we assessed whether the accumulation of PNGS in HA's globular head coincides with the emergence of non-PNGS mutations. To understand where the non-PNGS mutations appeared on HA after the addition of each glycan, we examined the non-PNGS mutations in the year where the glycan was added (Supplementary Data 1). First, we observed that the globular head is the region with the highest frequency of amino acid mutations. The mutations are mostly located in loops, outside secondary structure elements (Fig. 1E, panel I). Second, we detected that the amino acid mutations in the RBS are residues that are not directly involved in receptor binding (Fig. 1E, panel II). This is consistent with the fact that 130-loop, 150-loop, and 190 alpha-helix, which are essential for receptor binding, are relatively conserved among HA subtypes[7,10,22–24].

In summary, our analysis reveals a dynamic accumulation of PNGS on HA over the years, suggesting a discernible trend regarding the emergence and accumulation of PNGS in non-conserved regions of HA's globular head.

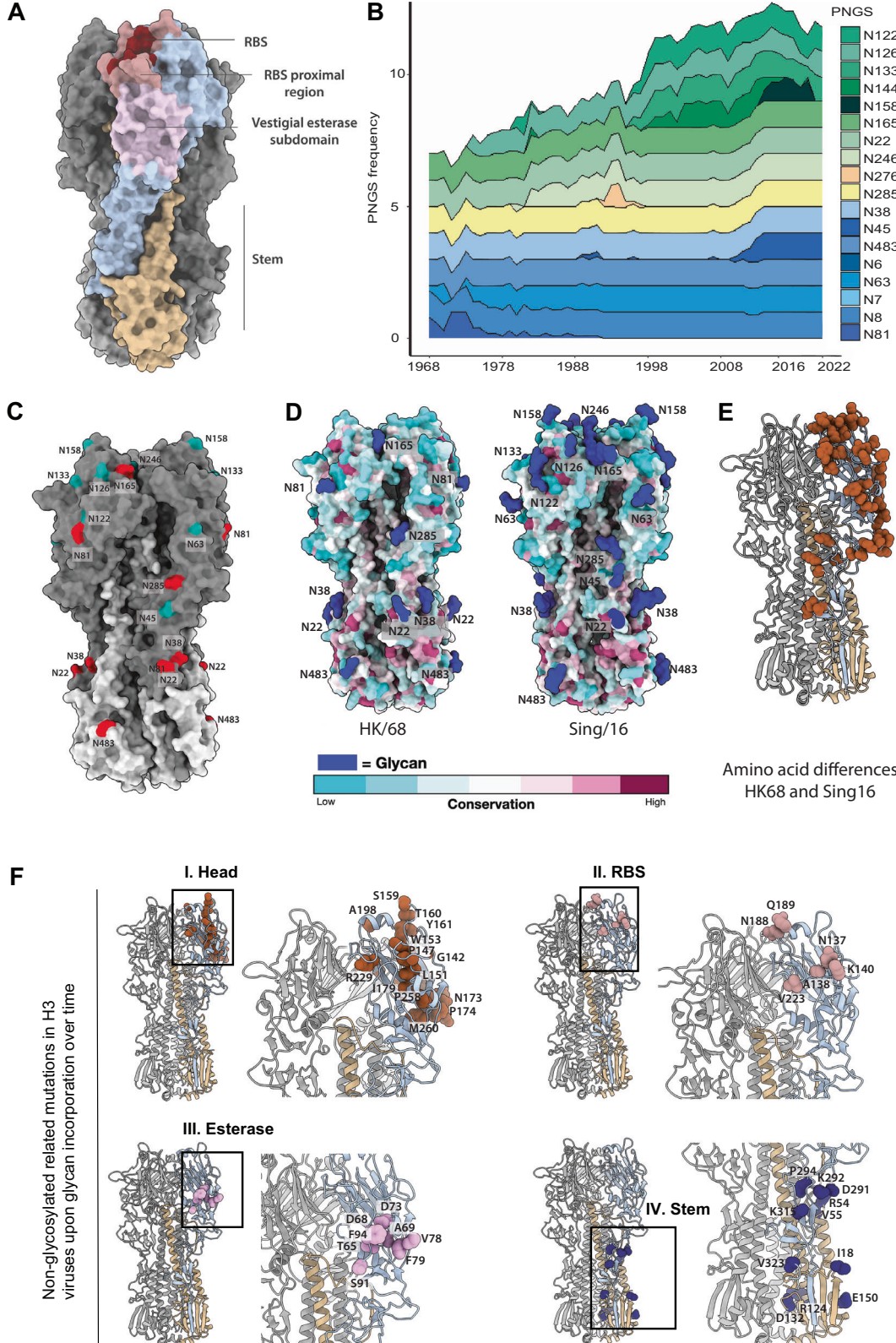

## Defining the structure and glycosylation profile of Sing/16

To better understand the complex relationship between glycan organization on the HA surface and its potential impact on emerging strains, we characterized the structures and glycosylation profile of both HK/68 and Sing/16 by cryo-EM (Fig. 2A–E and Supplementary Fig. S2) and mass-spectrometry (Fig. 2F, G). Notably, we obtained a structure of a fully glycosylated Sing/16 at 3.9 Å-resolution (Fig. 2A and

Supplementary Fig. S2). Both the HK/68 as the Sing/16 H3 proteins were purified from HEK293T as well as HEK293S GnTI- cells, in which N-glycans remain in a high mannose form as they lack the ability to undergo further processing into complex-type glycans. The HA plasmids contain either one or two N-terminal superfolder green fluorescent protein (sfGFP) domains, for HK/68 and Sing/16, respectively. The addition of the sfGFP increases the yield of the protein, can

**Fig. 1 | Computational analysis of PNGS and non-glycosylation related mutations in H3 viruses over time. A** Schematic representation highlighting HA domains: head (blue), RBS (burgundy), RBS proximal regions (salmon), vestigial esterase subdomain (pink) and stem (yellow). **B** Stacked graph showing the frequency of each PNGS within circulating strains from 1968 to 2022. This analysis includes - 11,000 sequences, which were downloaded from the influenza research database NCBI Virus[67]. Supplementary Table 1 contains the frequency of all PNGS over time. **C** HA structural domains represented on HK/68 (PDB ID: 4FNK). HK/68 glycans are highlighted in red and Sing/16 glycans in cyan. All residues on HA are named according to H3 consensus numbering A/Aichi/1968. **D** Residue conservation of HK/68 and Sing/16 using - 11,000 sequences from the influenza research database projected as a colormap on the HA structure. Glycans are highlighted in blue. **E** HA residues with distinct amino acid identities between HK/68 and Sing/16 H3N2 strains are depicted as brown spheres on a single protomer of the HK/68 HA trimer (PDB ID: 4FNK). **F** Mutations appearing in H3 viruses at the same time as the emergence of new PNGS. The different structural regions of HA are highlighted in brown, salmon, pink and blue for globular head, RBS, vestigial esterase and stem regions, respectively.

improve angular distribution in cryo-EM, and serve as a tool for immune tracing while not affecting antigenicity[25].

The structure of Sing/16 revealed the prototypic trimeric structure of HA (Fig. 2A and Supplementary Fig. S2). Each subunit consists of a membrane-distal polypeptide (HA1) and a membrane-proximal polypeptide (HA2). The HA1 subunit is comprised of a globular domain (residues 116–261) formed by an eight-stranded antiparallel β-sheet motif that includes a distal shallow pocket corresponding to the receptor-binding site, surrounded by the five immunodominant epitopes (A-E)[26]. The stem domain consists of a descending β-sheet motif from the HA1 subunit and is mainly formed by a helical coiled-coil structure corresponding to the HA2 subunit[26]. We also solved the cryo-EM structure of Sing/16 produced in GnTI- cells, at 3.7Å-resolution and Sing/16 produced in HEK293T, at 3.9 Å-resolution (Supplementary Fig. S2). The two structures are highly similar with a Cα-root-mean-square deviation (rmsd) of 0.77 Å (Supplementary Fig. S3), suggesting the distribution of glycan types does not have a significant impact on the folded structure. The cryo-EM structure of HK/68 was resolved to 3.4 Å-resolution (Supplementary Fig. S2) and is highly similar to Sing/16 (Ca-rmsd of 0.68 Å), demonstrating that the overall structure was maintained over time despite a relatively large number of mutations and the addition of PNGS (Fig. 2B–E).

We observe density for core N-acetylglucosamine (GlcNac) moieties in our cryo-EM reconstructions (Fig. 2B–E) at all PNGS on both HK/68 and Sing/16 except at N122, which is within a well-resolved region of the protein, and N8, which is near the N-terminus and is part of a poorly ordered region of the protein. Although we can observe density for individual glycans, we cannot identify the specific glycoforms and levels of occupancy from cryo-EM alone. We therefore employed glycoproteomics-based liquid chromatography–tandem mass spectrometry (LC-MS/MS) to characterize the site-specific distribution of glycoforms at each PNGS for both strains (Fig. 2F, G). While we identified glycopeptides covering most PNGS, we lack coverage at N22 and N81 on HK/68, and N63 on Sing/16, either due to low peptide detection or poor fragmentation. Of the glycans detected, however, we found that HK/68 possessed on average 67% complex, 15% hybrid, and 16% high-mannose type glycans, while Sing/16 possessed 49% complex, 9% hybrid, and 28% high-mannose. Of the PNGS that are shared between the two strains, the largest change occurred at N165, which went from 20% high-mannose on HK/68 to 70% on Sing/16, in line with previous results describing a progressive increase in glycan-mediated occlusion of the HA surface, particularly within the immunogenic globular head associated with the emergency of high-mannose glycans positions N165 and N246[27].

### Glycan accumulation on the HA head affects epitope exposure

To investigate how changes in HA glycosylation affect epitope exposure we first utilized a modified version (see "Methods" and Supplementary Fig. S9) of our previously developed high-throughput atomistic modeling pipeline to generate a large ensemble of fully glycosylated HA models of both HK/68 and Sing/16 (Fig. 3A)[28,29]. From these ensembles, we calculated the glycan encounter factor (GEF) across the solvent-exposed surface of HA, which is a measure of the number of glycan heavy (non-hydrogen) atoms encountered by an external probe approaching the protein surface (see "Methods"). We found that the accumulation of glycans on HA's head domain has a dramatic effect on the epitope accessibility in the region (Fig. 3B), with an average 4.7-fold increase in GEF for Sing/16 over HK/68 for head domain residues (Fig. 3B and Supplementary Fig. S4A, B). Normalized GEF values per residue, ranging from CYS52 to CYS277, were used for this calculation. In addition to the changes in the shielded surface area, we also observed an increase in average inter-glycan and glycan-protein contacts of glycans on the head domain of Sing/16 compared to the same glycans present on HK/68, which is a result of increased local crowding and glycan-glycan interactions (Supplementary Fig. S4B). This is important because the increased local contacts indicate reduced temporal dynamics of glycans within or around antibody epitopes can have an additional impact on epitope accessibility and antibody binding beyond just steric shielding effects[16].

Next, we measured binding of six broadly neutralizing antibodies (bnAbs) against the stem (CR9114)[30] and the RBS (S139/1[31], K03.28[32], TJ5-5[33], C05[34] and F045-92[35]) regions of HK/68 and Sing/16 (Fig. 4A and Supplementary Fig. S5A, B). HAs were expressed in HEK 293 T, HEK 293S GnTI- cells or HEK 293S GnTI- followed by treatment with endo H and antibody binding was measured by ELISA. The glycosylation state of HK/68 and Sing/16 from differential expression and endo H treatment did not significantly affect the $EC_{50}$ or AUCs of the stem or RBS antibodies (Fig. 4A and Supplementary Fig. S5A, B). Accordingly, after calculating the glycan encounter factor between HK/68, Sing/16 and C05, we observe that although the C05 epitope is surrounded by several acquired PNGSs on Sing/16, including N133, N158, N165, and N246, it does not directly clash with any of them (Supplementary Fig. S5C).

Finally, to evaluate how glycans impact the natural antibody response against HA, we analyzed reference sera from the Erasmus Surveillance Institute from ferrets immunized against A/Bilthoven/16190/68, A/NL/109/03 (NL/03) and Sing/16 HA by HA inhibition (HAI) assay. Briefly, ferrets were inoculated with the virus intranasally, and serum was collected two weeks later. Antibody titers were measured by HAI assay, which determines antibody titers against HA by measuring the dilution factor of serum required to inhibit the interaction between the viral HA and the sialic acid receptor present on red blood cells, measured by the ability of serum antibodies to inhibit hemagglutination[36]. NL/03 was included to understand antibody responses from a time point between 1968 and 2016.

The results indicate a higher inhibition titer of the ferret antisera raised against Sing/16 (10,240) compared to HK/68 (2560) and NL03 (160) (Supplementary Table S2). Similarly, our biolayer Interferometry (BLI) analysis demonstrated a greater magnitude of binding between Sing/16 and the corresponding sera compared to HK/68 and NL/03, suggesting that ferret inoculation with Sing/16 elicited a more robust immune response compared to inoculations with HK/68 and NL/03 (Supplementary Fig. S6A). These results may not necessarily reflect intrinsic differences between the HAs themselves but could instead be influenced by differences in the infections. Thus, we conducted a competition ELISA and an electron microscopy-based polyclonal epitope mapping (EMPEM)[37–40] assay to better understand how the

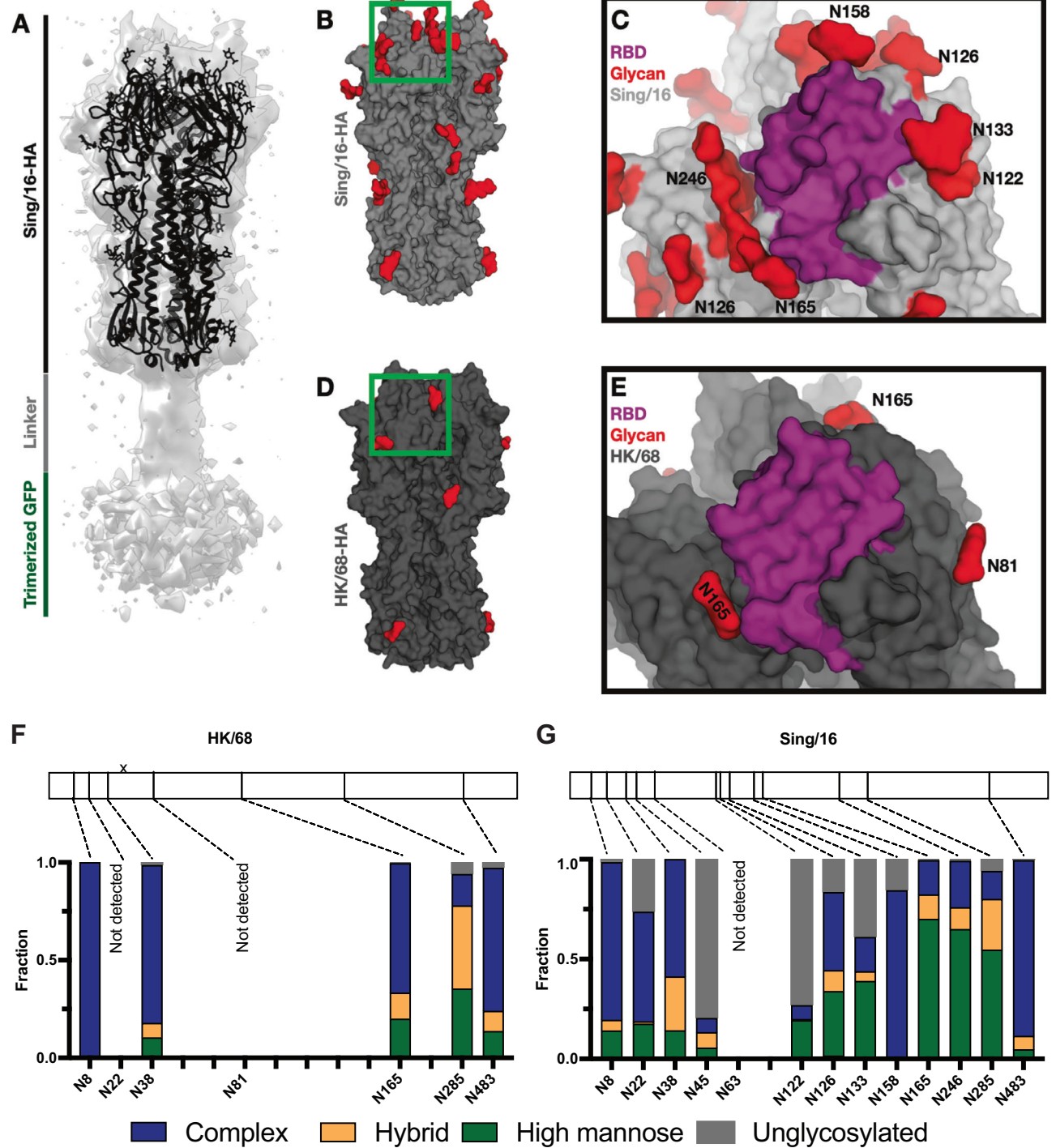

**Fig. 2 | Glycan shield composition and disposition of Sing/16 and HK/86 using cryo-EM and site-specific MS analysis showing changes over time. A** Cryo-EM structure of trimeric A/Sing/INFIMH/16 HA. Each monomer is composed of the globular HA1 domain and the stalk-like HA2 domain. HA2 is attached to 2 x sfGFP per monomer. Cryo-EM structures of Sing/16 (**A**, **D**, **E**) and HK/68 (**B**, **C**) with the glycans highlighted in red and a close-up view of the RBS region. **F**, **G** Site-specific MS analysis of HK/68 and Sing/16 glycan shields. Bar plots show the percentage of high-mannose (green), complex (blue), and unmodified peptides (gray) detected at each PNGS. At some sites, no peptides were detected. Source data are provided as a Source Data file.

polyclonal immune response elicited after immunization with the different strains differed from one another.

Our EMPEM analyses revealed that the fragment antigen binding domains (Fabs) of polyclonal antibodies complexed with matching proteins targeted the HA head in HK/68, head and vestigial esterase domain in NL/03, and vestigial esterase and stem domains in Sing/16 (Fig. 4B–D and Supplementary Fig. S6). Similarly, we observed

that while the presence of RBS bNAbs competed with polyclonal IgGs and Fabs isolated from sera in HK/68 and NL/03, only the stem bNAb CR9114 competed polyclonal IgGs and Fabs in Sing/16 (Fig. 4B, C and Supplementary Fig. S6). These data suggest that the increased number of glycans present on HA Sing/16 potentially favored the production of stem and esterase over head-directed antibodies against Sing/16.

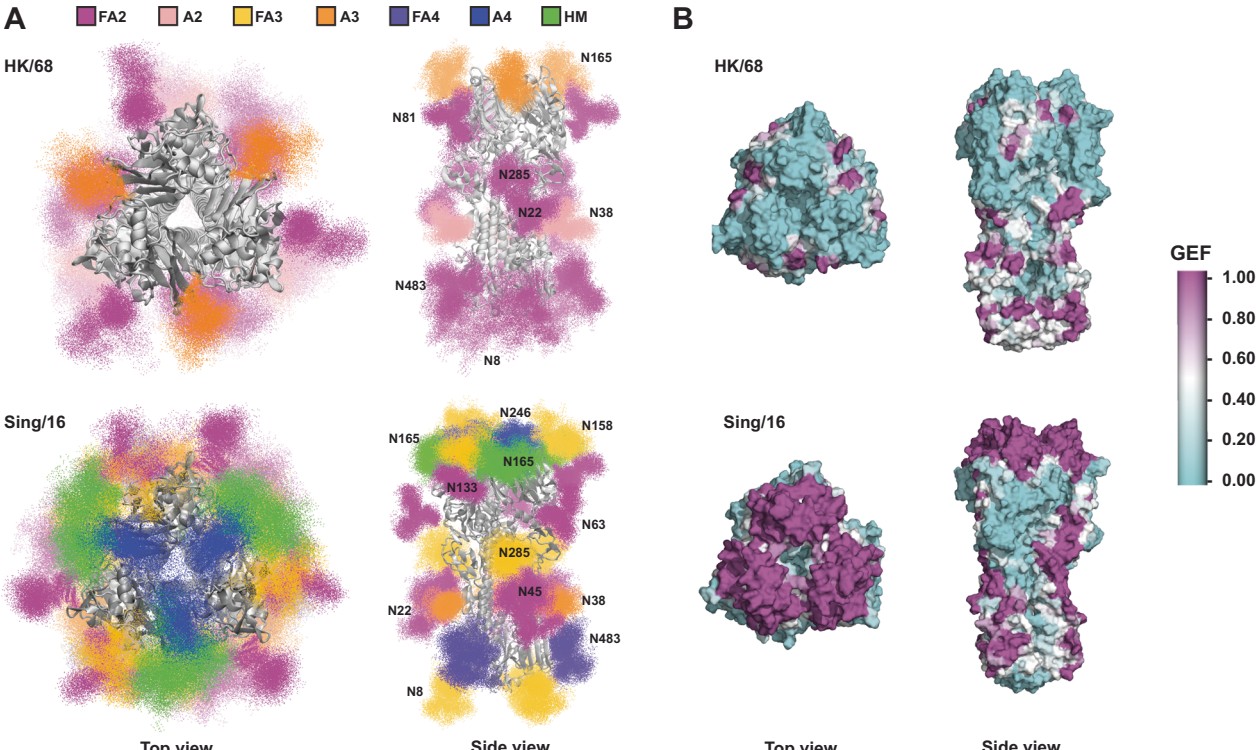

**Fig. 3 | Glycan shield analysis via high throughput ensemble modeling indicate increase of glycan surface area over time. A** Models of fully glycosylated HA trimers showing 20 poses for each glycan out of the 1000 relaxed glycan conformations generated via high throughput ensemble modeling. Glycan atoms are shown as points and colored by glycan type according to the legend. Glycan types indicate whether the glycan is fucosylated with an F and the number of branching antennae. High mannose glycans are indicated by HM. One occurrence of each glycan is labeled. Images are rendered with VMD 1.9.3. **B** Glycan encounter factor (see "Methods") mapped onto the surface representations of HK/68 and Sing/16. Magenta regions are considered shielded by glycans, while cyan regions are accessible.

## De-glycosylation destabilizes the HA head domain of Sing/16 but not HK/68

N-linked glycans can stabilize proteins through a variety of mechanisms[41,42], yet the biophysical consequences of HA glycosylation have not been thoroughly investigated. We previously showed that enzymatic deglycosylation of the Human Immunodeficiency Virus type-1 (HIV-1) Envelope glycoprotein leads to progressive destabilization and unfolding of the gp120 subunit, with the instability nucleating within the hypervariable loops 1-3 that compose the trimer apex[28]. Considering this, we sought to determine if HA was also being destabilized by endo H treatment. Therefore, we collected cryo-EM data of both endo H treated HK/68 and Sing/16 (Fig. 5 and Supplementary Fig. S7). Indeed, we observed significant destabilization of Sing/16 but not HK/68 following endo H treatment as seen from the cryo-EM 3-D classification results (Fig. 5A). Specifically, we observed that ~ 57% of the trimeric Sing/16 HA particles retained after 2-D classification exhibited some level of protein unfolding/destabilization primarily in the hyper-glycosylated head domain and to a lesser extent in the stem/membrane proximal regions (Fig. 5A). By pooling the particles from the well-folded 3-D classes (100% of the HK/68 particles) we obtained 2.3 Å and 3.2 Å-resolution reconstructions of endo H treated HK/68 and Sing/16, respectively (Fig. 5B). Both structures were nearly identical to their natively glycosylated counterparts in the protein backbone (Supplementary Fig. S3). endo H treatment of HK/68 resulted in a similar uniform distribution of views compared to endo H treated Sing/16 (Supplementary Fig. S2D). It is noteworthy that the resolution of 2.3 Å achieved for HK/68 represents the highest cryo-EM resolution reconstruction of HA to date, perhaps owing to the lack of glycans and isotropic tumbling in the vitreous ice on the cryo-EM grid. Difference

maps calculated by subtracting the aligned endo H treated maps from the GnT1- maps reveal the full extent of the glycosylation on both structures (Fig. 5C), highlighting the nearly complete shielding of the head domain on Sing/16. The extent of glycan density observed in the difference maps closely matched the glycan density observed in simulated density maps generated from the ensemble of glycosylated atomistic models shown in Fig. 3 (Supplementary Fig. S8), further validating the accuracy of the modeling procedure and confirming that the density features observed in the difference maps indeed arise from glycan signal. Interestingly, by comparing the local map intensity around individual glycans between the GnTI- and stable endo H treated 3-D reconstructions we found that several glycans were incompletely digested by endo H (Fig. 5D and Supplementary Fig. S7), most prominently the glycan at N165, which retained ~10% and ~ 50% occupancy on HK/68 and Sing/16, respectively (Fig. 5E). In addition, 6 other glycans on Sing/16 also showed incomplete de-glycosylation, specifically, N8, N45, N63, N133, N246, and N285 (Fig. 5E). Finally, we measured the thermal stability of both HAs before and after endo H treatment via differential scanning fluorimetry (DSF) and observed a marked reduction in melting temperature of Sing/16 relative to the fully glycosylated sample along with the appearance of a second apparent melting transition (Fig. 5F and Supplementary Fig. S6).

## Discussion

The intricate interplay between HA glycosylation and the host immune response is a central factor influencing the pathogenesis of IAV and the emergence of viral strains. In our assessment of how the presence of glycans impacts protein heterogeneity, we observed notable changes in the dynamics of glycans within or

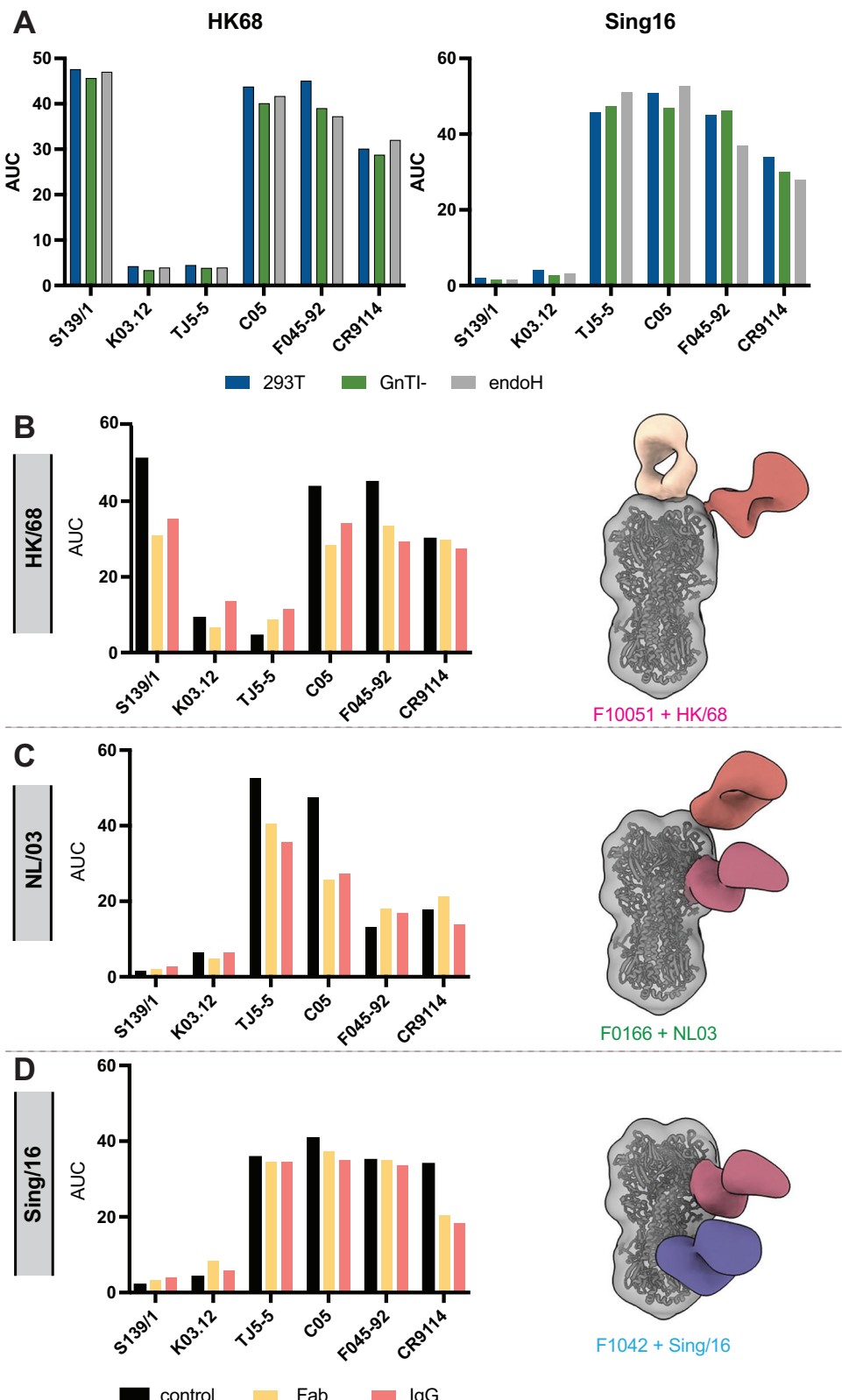

near antibody epitopes. The increased local crowding and interactions among glycans suggest a potential mechanism influencing antibody recognition and binding (Figs. 1 and 3). These changes, alongside alterations in glycan shielded surface area, can influence epitope accessibility and antibody binding, extending our understanding of glycans beyond their traditional role as steric hindrances.

Of the glycans that appeared on Sing/16 globular head, the appearance of the PNGS at N158 has been previously discussed[12,19,43]. The complex-type fraction of the PNGS at N158 has been to be a major obstacle for α2-6 receptor binding[18,19,44]. Interestingly, recently, the NYT158-160 sequon has been replaced with NNI in currently circulating H3N2 viruses, restoring some α2-6 receptor binding. Of note, the appearance of N158 has been proposed as the most likely cause of

**Fig. 4 | Immunogenicity analysis of HK/68 and Sing/16 display changes in the epitopic landscapes of HA. A** RBS-specific bnAbs (S139/1, K03.12, TH5-5, C05 and F045-92) and stem-specific bnAb CR9114 binding titers to HA measured by ELISA. Area under the curve (AUC) is shown in bar graphs. HK/68 and Sing/16 HA proteins were purified from HEK 293 T, HEK GnTI- cells, and HEK GnTI- cells followed by treatment with endo H. Experiments were performed in duplicate. Error bars are shown. **B** Competition ELISA using purified polyclonal IgG and polyclonal Fab from reference ferret sera was used to measure RBS-specific and stem-specific binding titers to HK/68 (**B**), NL03 (**C**), and Sing/16 (**D**) by ELISA. Experiments were performed in triplicate, and AUCs are shown in bar graphs. Composite 3-D reconstructions from EMPEM analysis of polyclonal Fabs were obtained from reference ferret sera (F10051, F0166 and F10421) and complexed to matching HA glycoprotein (HK/68, NL/03 and Sing/16) purified from HEK 293 T cells. EMPEM composites were made using Chimera X version 1.8 and are shown on the right panels of (**B**, **C** and **D**). Experiments were performed in duplicate. Error bars are shown. Source data are provided as a Source Data file.

antigenic distancing between different influenza strains circulating between 2012 and 2018[43].

Previously published studies have proposed that the N158K substitution on influenza egg-based vaccines contributes to the loss of the glycan at N158, leading to vaccine antigenic mismatch[21]. According to our computational analysis, PNGS at N158 appears on circulating strains between 2014 and 2020 but then disappears in 2021/2022 (Fig. 1A and Supplementary Fig. S1). Interestingly, the yearly CDC report highlights that, despite inadequate vaccine enrollment to generate dependable vaccine effectiveness estimates by age group or vaccine type, immunization (2021/2022) did not diminish the risk of outpatient medically attended illness associated with influenza A(H3N2) viruses[45]. This highlights the significance of glycans when considering vaccine design and manufacturing methods.

We show that Sing/16 ferret-specific sera elicited an increased neutralizing effect when compared to HK/68 or NL/03. Notably, EMPEM and competition ELISA reveals a distinctive pattern: antibodies against HK/68 and NL/03 primarily targeted the head and the vestigial esterase of the HA, while antibodies against Sing/16 were directed towards the esterase and stem regions (Fig. 4). This is in line with our data showing that glycans did not impact the effectiveness of stem directed bnAb when comparing HK68 and Sing/16 (Fig. 4). These findings support growing perspective that vestigial esterase and stem-targeting antibodies can provide effective viral neutralization[46,47].

Building on the observed effects of glycans on antibody binding, our cryo-EM data provide additional insight into their structural contributions to HA stability. The incomplete de-glycosylation observed after endo H digestion (Fig. 5D) has two implications; first, that the increased glycan density in the HA head restricts endo H access; and second, that the partially unfolded 3-D classes likely represent a subset of particles where these glycans were successfully cleaved, or conversely, that the glycans with partial occupancy in the well-folded trimer classes are particularly important for H3 stability. For instance, the glycan at N165, which shows the highest apparent occupancy in both endo H treated cryo-EM maps, is also the most ordered as measured by local map intensity (Fig. 5D). It is important to note that endo H cleaves oligomannose glycans after the core GlcNac residue, thus the instability cannot be explained by the loss of local glycan-protein interactions which predominantly involve the core GlcNAc.

N165 is the only conserved PNGS present on HA's head shared between Sing/16 and HK/68. However, N165 switches from being mainly complex in HK/68 to being mainly HM in Sing/16. In fact, 3 out of 5 of the analyzed glycosides appearing on HA's head/RBS region (N126, N165, N246) present an increased HM fraction when compared to HK/68 (Fig. 2A, B). The intricate mechanisms through which clustered N-glycan sites foster high mannose species were previously discussed[48]. Essentially, this phenomenon is believed to be driven by the presence of a high density of glycans that sterically constrain the α-mannosidase-mediated glycan trimming during Golgi glycan processing. This pattern holds true across different viruses, particularly evident in HIV, where it is consistently observed across distinct clades and production systems[49]. Finally, the differential impact of deglycosylation on Sing/16 stability relative to HK/68 may indicate that the gradual accumulation of N-linked glycans in the HA head over time have co-evolved with non-PNGS mutations that would otherwise be

detrimental to HA stability and therefore viral fitness in the absence of these glycans. These deleterious mutations could be within the PNGS sequons themselves or elsewhere in the protein, such as within the more conserved domains known to be important for HA stability.

In summary, our findings underscore the intricate nature of glycan-protein interactions and their significant role in modulating HA stability and immune escape, both directly by blocking access to neutralizing epitopes but also allowing additional mutation of epitopes on the HA surface. Understanding the structural and biophysical interplay between HA and glycans can be used in the design of better and more stable vaccine candidates.

## Methods

### Code glycan prevalence over time

11,187 HA amino acid sequences were pairwise aligned to the reference sequence CAA24209 A/Aichi/2/1968 and analyzed using the Anchor tool package[50]. Alignments were grouped by year, and for each year each glycan position was queried with the pattern NXT/S, representing an asparagine followed by a threonine or serine residue at position N + 2. Positions 22, 23, 24, 38, 54, 61, 79, 97, 138, 142, 149, 160, 174, 181, 262, 292, 301, and 499 counted from the CAA24209 reference were queried for the year range 1968–2022 for all sequence data. Counts and totals for each matching sequence at each position were collated and analyzed to determine prevalence.

### Protein Expression and purification

pCD5-HA-GCN4-Fluorescent probe expression vectors were transfected into both HEK293T and HEK293S GnTI(-) cells (which are modified HEK293S cells lacking glucosaminyl transferase I activity (ATCC CRL-3022) with polyethyleneimine I (PEI) in a 1:8 ratio (μg DNA:μg PEI) as previously described[25]. The transfection mix was replaced after 6 hours with 293 SFM II suspension medium (Invitrogen, 11686029, supplemented with glucose 2.0 g/L, sodium bicarbonate 3.6 g/L, primatone 3.0 g/L (Kerry), 1% glutaMAX (Gibco), 1.5% DMSO and 2 mM valproic acid). Culture supernatants were harvested 5 days post-transfection. The HA expression was analyzed with SDS-PAGE followed by Western blot on PVDF membrane (Biorad) using α-strep-tag mouse antibodies 1:3000 (IBA Life Sciences). Additionally, fluorescence intensities were measured using a filter based PolarStar Omega plate reader. Subsequently, HA proteins were purified with Sepharose Strep-Tactin beads (IBA Life Sciences).

### ELISA and competition ELISA

HK/68 and Sing/16 were HEK293T and GnTI-derived. GnTI-derived protein was further treated with Endo H (1000 units) for 5 h for the removal of high-mannose structures. Proteins were coated on maxisorp plates (Invitrogen) at a concentration of 2.5 μg/mL in PBS, overnight at 4 °C. Blocking was performed for 2 h using 3% BSA, followed by incubation with primary antibody at 40 μg/mL for 2 h at room temperature. Following the primary antibody, a secondary antibody with an HRP conjugate (1:2000) was added, with an incubation of 45 min at room temperature. Finally, the plates were developed with TMB substrate solution (34028, Thermo Scientific) and the reaction was stopped after 3 min using 2.5 M $H_2SO_4$. The competition ELISA was performed similarly, but after the blocking step and before adding the

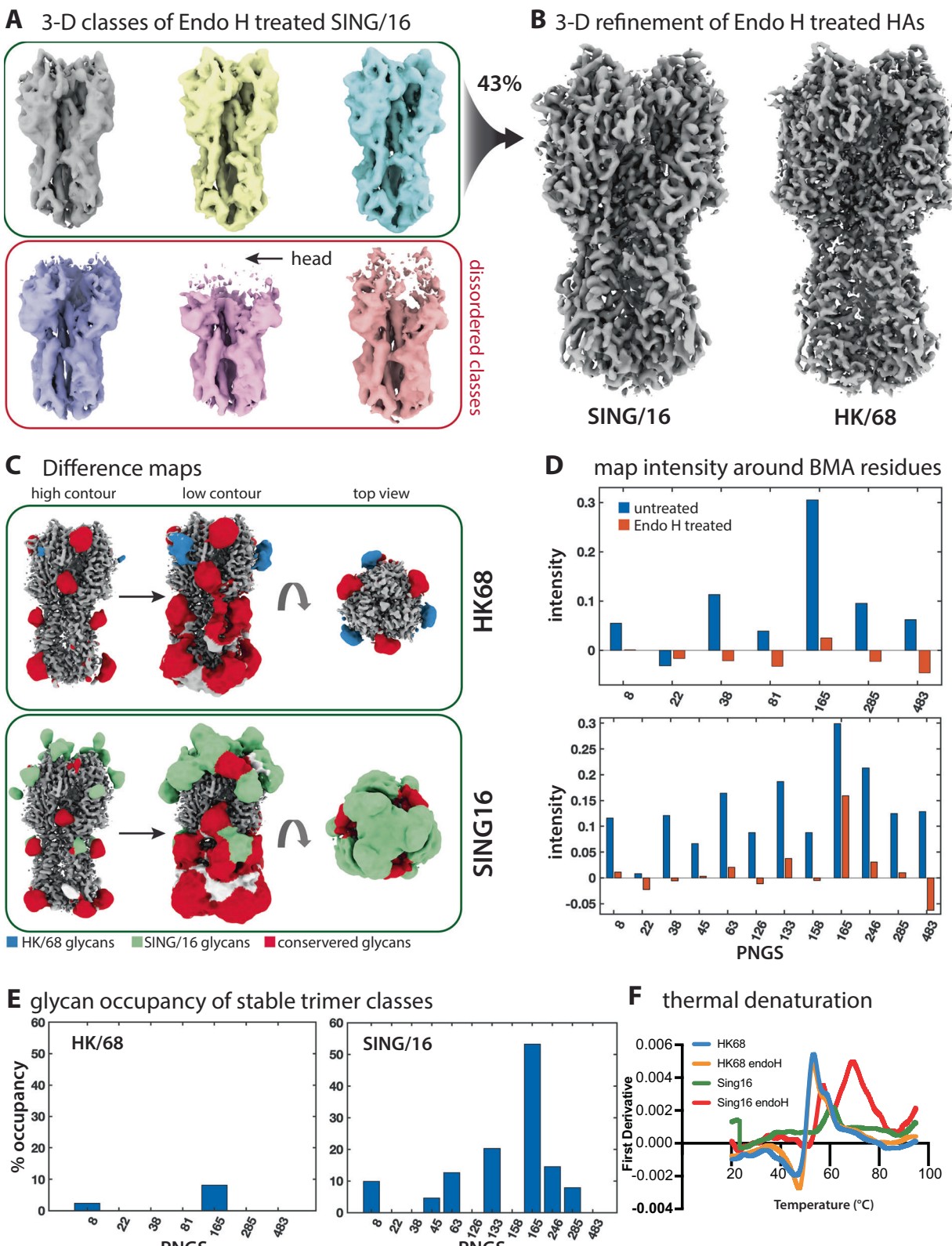

**Fig. 5 | Enzymatic deglycosylation by endo H treatment destabilizing Sing/16 HA. A** Cryo-EM 3-D classification results of HK/68 and Sing/16 HA after treatment with endo H, and (**B**) the resulting 3-D refinement of the selected and pooled classes. Also shown are percentages of retained and discarded particles and an overlay of 3-D reconstructions representing the two extremes of the 1st principal component from 3-D variability analysis in CryoSparc. **C** Glycan difference map of HK/68 and Sing/16. **D** Normalized glycan mean intensity for untreated HA (blue) and endo-H treated HA (orange). **E** Glycan occupancy around the third glycan residue, β-mannose (BMA) between the GnTI- and stable endo H treated 3-D reconstructions of HK/68 and Sing/16. **F** Thermostability of both HA variants obtained from nano-DSF measurements. Data was plotted using ChimeraX and Prism 9. Source data are provided as a Source Data file.

primary antibody, we incubated polyclonal Fab or IgG isolated from ferret sera at a concentration of 20 µg/µL in all wells.

## Endoglycosidase H digestion of HK/68 and Sing/16 for cryo-EM experiments

0.5 mg of purified HK/68 or Sing/16 from HEK293S GnTI- cells was mixed with 20,000 units of Endoglycosidase H (Endo H; New England Biolabs) in non-denaturing reaction buffer to a final volume of 0.5 ml and incubated at 37 °C for 5 h. To quench the reaction and purify the sample for cryo-EM experiments, the HAs were run over a size exclusion column Superdex 200 increase (Cytiva Life Sciences), and fractions were collected and concentrated.

## Ferret samples used in the study

For the EMPEM assays used in this paper, serum samples for ferrets F10051, F0166 and F1042 obtained from Erasmus Surveillance Institute were used to test HK/68, NL/03 and Sing/16, respectively.

## BLI/OCTET

Biolayer interferometry assays were performed using the Octet Red96 instrument (Sartorius, ForteBio). IgG from serum was immobilized onto Protein A Biosensors for 120 s followed by a 60 s baseline measurement in Kinetics buffer (PBS pH 7.2 with 0.01% w/v bovine serum albumin and 0.002% v/v tween-20). The biosensors were then dipped for 300 s into wells containing HK/68, NL03 or Sing/16 diluted in Kinetics buffer at a final concentration of 200 nM. The sensors were dipped into wells containing Kinetics buffer for 600 s to assess dissociation.

## Serum IgG isolation and antibody digestion

IgG from ferret sera was isolated using CaptureSelect IgG-Fc Affinity Matrix (Thermo Scientific) as previously described[40,51] In summary, 0.5 ml of ferret sera was mixed with 0.5 ml of washed CaptureSelect resin and 4 ml of PBS. For IgG digestion, the unbound IgG was discarded, and the resin was kept. Papain was activated for 15 min at 37 °C in digestion buffer (100 mM Tris, 2 mM EDTA, 10 mM L-Cysteine, 1 mg/ml papain) and was added to the resin containing the IgG. Subsequently, digestion buffer (20 mM sodium phosphate, 10 mM EDTA, 20 mM cysteine, 0.1 mg/ml papain, pH 7.4) was added to the resin up to a total volume of 5 ml and the mixture was incubated for 4-5 h at 37 °C. Iodacetamide was used to quench the reaction at a final concentration of 0.03 M, and the Fab/Fc from digested IgG was purified and concentrated using the size exclusion chromatography (Superdex 200 increase column, Cytiva Life Sciences). The fractions containing purified Fabs/Fc were concentrated using 10 kDa Amicon ultrafiltration units.

## Purification of antigen-Fab complexes

Antigen-Fab complexes were generated by two methods: first, 15 µg of antigen were incubated with 1 mg of Fab overnight at RT. The complexes were purified using a Superdex 200 increase column on Akta Pure System (GE Healthcare) running in TBS buffer. The fractions containing the complexes were concentrated using 10 kDa Amicon ultrafiltration units and immediately added to a negative-stain grid. Second, immune complexes were prepared in a molar proportion of ~1:10 (antigen:polyclonal Fab), using 2 µg of antigen and 60 µg of polyclonal fab incubated overnight at room temperature. The unbound Fab was washed using TBS in a 100 kDa Amicon Ultra 0.5 mL centrifugal filter (Merck Millipore). Fab-antigen complexes were concentrated in the Amicon Ultra until they reached a concentration of 20 µg/ml.

## Electron microscopy

**Negative-stain sample preparation and imaging.** Fab-antigen complexes were diluted to 20 µg/ml and applied for 10 s to 400 mesh Cu grids that were carbon coated and glow discharged at 15 mA for 25 s. The Fab-antigen complex was negatively stained with 2% uranyl-formate for 50 s. Data was collected using a Tecnai Spirit electron microscope at a nominal magnification of 52,000 X with a pixel size of 2.06 Å. The defocus range was set between −1.5 and −2 mm and the electron dose was 25 e⁻/Å². Micrographs were recorded using a Tietz (4k) TemCam-F416 CMOS, and data was acquired using the automated imaging interface from Leginon[52].

**Negative-stain data processing.** For antigen-Fab complexes, ~150,000 particles were picked using the Appion image processing package. Particles were transferred to Relion/3.0, and 2-D classification was performed[53]. Particles that contained trimer only or trimer-Fab complexes were selected for 3-D analysis. The 3-D reference for 3-D classifications and refinements was a low-resolution model of a non-liganded HA. Initial 3-D refinement was performed using a minimum of 100,000 particles to align all the particles prior to 3-D classification. Particles were then classified into 20–40 classes, and classes with similar features were combined and refined. Particles were transferred to Relion/3.,0 and 2-D classification was performed[53].

**Cryo-EM sample preparation.** HK/68 and Sing/16 were diluted to a final concentration of 1 mg/ml and mixed with 0.5% Lauryl maltose neopentyl glycol (LMNG). Sing/16 was diluted to a final concentration of 7 mg/ml, 1 mg/ml and 0.2 mg/ml and mixed with 4 mM CHAPSO, 0.5% LMNG and 0.3% octyl beta glucoside (OBG), when expressed in HEK293T, HEK293S GnTI- and Endo H-treated, respectively. UltrAufoil R 1.2/1.3 300 mesh gold grids were plasma cleaned for 20 s at 15 mA using the Solarus advanced plasma cleaning system (Gatan) before loading the sample. Further, 3 µL of the sample was loaded onto the grid and plunge-frozen in nitrogen-cooled liquid ethane using the Vitrobot Mark IV (Thermo Fisher Scientific). The settings for the Vitrobot were as follows: Temperature: 4 °C, humidity: 100%, blotting time: 5.5 s, blotting force: 1 and wait time: 3.5 s.

**Cryo-EM data collection and image processing.** Grids were loaded into a Titan Krios Cryo-Transmission Electron Microscope (FEI) operating at 300 kV and equipped with a K2 Summit direct electron detector camera (Gatan). The data was collected at a total cumulative dose of 50 e⁻/Å². Magnification was set at 36,000x with a resulting pixel size of 1.15 Å/pix at the specimen plane. Automated data collection was performed using Leginon software[52]. The data collection details are presented in Supplementary Table S1.

The micrograph movie frames were aligned and dose-weighted using MotionCor2[54]. In cryoSPARC v3.1.0[55], patch CTF was applied, and micrographs were manually curated. Further, particles were picked from micrographs using blob picker and template picker. The particles were then extracted, and a few rounds of 2-D classification were performed. Particle picks were subjected to a round of 3D classification and Non-Uniform 3-D Refinement. The highest resolution classes were selected and refined in 3-D refined while imposing C3 symmetry. In Chimera, the HA trimer A/Hong Kong/1/1968 (PDB entry 4FNK) crystal structure was aligned with HK/68 or Sing/16 and used as the initial model.

**Model building and refinement.** The final sharpened maps were used for subsequent model building. Multiple rounds of Rosetta relax refinement[56] and manual Coot refinement[57] were performed. To validate the analysis, EMRinger[58] and MolProbity[59] were used. The final refined model was submitted to the Protein Data Bank (PDB). Structural figures were generated using UCSF Chimera X[60] or Pymol[61].

## Computational modeling

**Screening and elimination of glycan burial artifacts in models.** Often, a molecular dynamics (MD) simulation produces a trajectory

showing a glycan position that is, while thermodynamically stable, not realistically observed in experimentally derived models of the same glycoprotein[29,62,63]. This can cause glycans to be improperly buried inside the protein scaffold. Since recalculating the dynamics of a glycoprotein can be time consuming, a method of quickly determining the possible burial of a glycan within the protein scaffold was needed to help discern which glycosylation site on the scaffold would, in the case of the glycan being buried, need to be structurally exposed, via homology modeling, to obtain a realistic (unburied) spatial glycan position. Two separate computational methods were developed[28,29] and tested on 3 different viral spike glycoproteins belonging to three different viruses (HIV: BG505 SOSIP SIV: gp120/gp41 complex and Human Influenza: Hemagglutinin A (HA)). Here, HIV SOSIP Envs have been used as a test system for influenza since this is an update to our previous modeling method initially designed on SOSIPs[28,29].

Initially, it was hypothesized that measuring the Solvent Accessible Surface Area, or SASA, as a metric of the degree of glycosylation site (NGS) exposure would be a quick and computationally efficient method of predicting the possibility of glycan burial, given that this site is later glycosylated without the need for the glycan to be present on the structure at the time of analysis. In theory, performing this for every NGS of each of the five viral glycoproteins would give two approximate ranges of SASA values separated by a central cutoff methods value over which an NGS's position would or would not result in a buried glycan. A script was written in Python-3[64] in conjunction with the Biopython[65] module for Python-3 to parse each unglycosylated protein structure in Protein Data Bank (PDB) format. A Shrake-Rupley algorithm[66] for calculating SASA is then performed on the structure, and the SASA value of each protein residue with respect to its adjacent residues is appended to its metadata in the structure object. Examples of the resulting SASA of each potential NGS and control non-glycosylation site asparagine residues of HA and BG505 are plotted as shown in Supplemental Fig. 7 panels A and B. Although cutoff values could be approximated as the threshold between prospective buried and unburied glycans somewhat clearly in the case of HA, no correlation could be determined between SASA and glycan burial for more heavily glycosylated proteins like BG505 and gp120/gp41, as shown in Supplemental Fig. 7 panel C. For this reason, a new method was developed to better classify glycan burial in a more robust, yet computationally taxing, manner.

As it was observed that most instances of glycan burial occurred at the tips of the structure and not the glycan 'stalk' to which the NGS is bound, a new method was developed in python to iterate over every residue in every glycan of a structure and calculate the distances from each glycan residue to the center of the protein scaffold and the carbohydrate residue's closest surface point to the center of the protein scaffold, comparing these two values, and using this comparison to determine glycan burial; the logical details of this method are described further via the pictorial algorithmic flow chart provided in Supplemental Fig. 7 panel J. The burial depth in angstroms of each sugar residue of each glycan is then plotted on a heat map for each structure to quickly determine each residue's relative degree of burial, examples of glycan positions and their corresponding heat maps can be seen for BG505 in Supplemental Fig. 7 panels D−I. Finally, an overall comparison between the entirety of the glycan shields of human influenza and HIV spike proteins can be seen in Supplemental Fig. 7 panels K and L, highlighting the much larger degree of glycan burial on HIV BG505. Our updated method has helped ensure that no glycans have been buried within folds of the underlying protein for the HA models used in this study.

**Modeling the glycan ensemble.** Based on PNGS sequence and cryo-EM data, the glycosylation sites present on the HK/68 and SING/16 variants were determined. MS was used to select the most probable glycoform at each PNGS when possible. For glycosylation sites where the structure could not be experimentally determined, the most common glycan type for these variants was assumed (fucosylated 2-antennae, FA2). Following the methods described earlier[28,29], an ensemble of 3D conformations of the glycosylated hemagglutinins was generated in atomistic detail. For each hemagglutinin variant, 10 protein structures determined via cryo-EM were used as templates for glycosylation. Glycans were added to the known glycosylation sites with ideal geometries as determined by the CHARMM36 force field then randomized by 1 Å across atomic coordinates. Following this randomization step, the glycan structures were relaxed via 1000 steps of conjugate gradient minimization. At this stage, the structures were equilibrated with a 500 ps molecular dynamic simulation. To ensure the conformational space was adequately sampled, five rounds of simulated annealing were performed in which the PNGS asparagine residues, loops, and glycans were unrestrained. For each of the 10 template structures, 100 glycosylated conformations were sampled. The protein backbones of these conformations were aligned to yield an ensemble of 1000 possible glycan orientations, which displays the glycan surface coverage.

**Glycan root mean square fluctuations.** To demonstrate the dynamics of each glycan, the root mean square fluctuations (RMSF) at three different positions were determined. These positions were the tip (C5 of the sugar furthest from the base), the core (C5 of the β-mannose), and the base (Cα of the glycosylated asparagine). The RMSF measures the conformational fluctuations across the 1000 poses and is averaged over the three protomers present in each hemagglutinin. Calculating the RMSF of each of these positions allowed for the comparison of the fluctuations of the glycan branches and stems, as well as the isolation of any backbone fluctuations.

**Glycan encounter factor.** The per-residue glycan encounter factor (GEF)[28,29] was calculated for each variant to quantify the extent of which the glycans effectively shield the surface of the underlying protein from external probes, such as antibodies. To do so, the geometric mean of the number of heavy (non-hydrogen) atoms encountered by a cylindrical probe with a 6 Å diameter approaching each residue of the proteins from the three Euclidean axes was averaged over the three protomers. A 6 Å probe was used to approximate the size of a hairpin loop – mimicking the initial encounter contact by a typical antibody. A geometric mean was performed over the x, y, z directions to ensure a residue that is accessible from one direction, was considered unshielded. The GEF was normalized by a factor of 1.5 (approximately $p = 0.05$), and any values greater than one were set equal to one to indicate shielding of that residue. A GEF of zero indicates no shielding of that residue from external probes. A surface representation of the two variants was colored by the GEF to show regions of glycan coverage.

**Simulated Cryo-EM Volumes from Atomistic Ensembles.** The simulated cryo-EM volumes (Supplementary Fig. S8) were generated with a custom Python script available at: https://github.com/ZTBioPhysics/Synthetic_Cryo-EM_Volume_Generator.git.

This script generates synthetic cryo-EM volumes from a set of PDB files. Each PDB file is read to generate a 3D density volume by depositing a precomputed Gaussian kernel at the location of every atom. The script can optionally filter out non-protein atoms (e.g., include only standard amino acids), and simulate experimental imperfections such as alignment errors (random translations and rotations) and additive Gaussian noise. Finally, the script sums the individual volumes and computes an average volume, which is then saved in MRC format for further visualization (e.g., in ChimeraX). For coherent results, the PDB files need to be pre-aligned to a common reference.

## Glycoproteomics analysis

**Glycoproteomics sample preparation.** For N-linked glycan analysis, the samples were denatured at 95 °C in a final concentration of 2% sodium deoxycholate (SDC), 200 mM Tris/HCl, 10 mM tris(2-carboxyethyl)phosphine, pH 8.0 for 10 min followed with 30 min reduction at 37 °C. Samples were next alkylated by adding 40 mM iodoacetamide and incubated in the dark at room temperature for 45 min. Samples were split in four for parallel digestion with trypsin (Promega), chymotrypsin (Roche), lys-C (Sigma), gluC (Sigma), and trypsin (Sigma). 2 µg sample was used for each protease digestion. For each protease digestion, the denatured, reduced, and alkylated samples was diluted in a total volume of 100 µL 50 mM ammonium bicarbonate, adding proteases in a 1:50 ratio (w:w) for incubation overnight at 37 °C. After overnight digestion, SDC was removed through precipitation by adding 2 µL trifluoroacetic acid (TFA) and centrifugation at 14,000 × g for 20 min. Following centrifugation, the supernatant containing the peptides was collected for desalting on a 30 µm Oasis HLB 96-well plate (Waters). The Oasis HLB sorbent was activated with 100% acetonitrile and subsequently equilibrated with 0.1% TFA in water. Next, peptides were bound to the sorbent, washed twice with 0.1% TFA in water and eluted with 40 µL of 50% acetonitrile/0.1% TFA in water (v/v). The eluted peptides were vacuum-dried and resuspended in 100 µL of 0.1% TFA in water.

**Glycoproteomics LC-MS/MS measurements.** For each sample and protease digestion, approximately 200 ng of peptides were analyzed by online reversed-phase chromatography using an Agilent 1290 UHPLC system coupled to a Thermo Scientific Orbitrap Eclipse Tribrid mass spectrometer. Peptide separation was performed using a Poroshell 120 EC C18 (50 cm × 75 µm, 2.7 µm, Agilent Technologies) analytical column and a ReproSil-Pur C18 (2 cm × 100 µm, 3 µm, Dr. Maisch) trap column. Samples were eluted over a 90 min gradient ranging from 0 to 55% acetonitrile. Peptides were analyzed at a resolution of 120,000 in MS1. MS1 scans were acquired with an AGC target set to 4e5, a maximum injection time of 50 ms, and a scan range of m/z 350−2000. Precursors were selected using a 1.6 m/z isolation window and fragmented by higher-energy collisional dissociation (HCD) with a normalized collision energy of 28%. HCD MS2 spectra were recorded at a resolution of 30,000, using centroid data mode and a scan range of m/z 120−4000. Oxonium ion detection was used to trigger additional fragmentation of glycopeptides. Upon detection of diagnostic glycan fragment ions, an additional scan of the same precursor was acquired using electron-transfer/higher-energy collisional dissociation (EThcD) with calibrated, charge-dependent ETD parameters and 27% normalized collision energy for supplemental activation. EThcD MS2 spectra were acquired in the Orbitrap at a resolution of 60,000 with a scan range of m/z 120−4000.

**Glycoproteomics data analysis.** One set of four parallel protease digestions (trypsin, gluC, chymotrypsin, lysC) was defined as a replicate analysis. These digestions were performed in duplicate, and each analyzed in technical triplicate for a total of six replicate analyses for HK/68 and Sing/16. The acquired data was analyzed using Byonic (v5.9.125) against a custom database of recombinant influenza HA protein sequences and the proteases used in the experiment, searching for glycan modifications with 10/20 ppm search windows for MS1/MS2, respectively. Up to 4 missed cleavages were permitted using C-terminal cleavage at R/K for trypsin, 6 at E/D for gluC, 6 at WFLYM for chymotrypsin or 4 at K for lysC. For N-linked analysis, carbamidomethylation of cysteine was set as fixed modification, oxidation of methionine/tryptophan as variable rare 2. N-glycan modifications were set as variable common 2, allowing up to max. 2 variable common and 2 rare modifications per peptide. All N-linked glycan databases from Byonic were merged into a single non-redundant list to be included in the database search. The Byonic result files were filtered for glycopeptides with scores ≥ 200, and additional peptide-spectrum matches (PSMs) containing non-glycosylated asparagines at PNGS were included at the same score cutoff. Glycans were classified based on HexNAc content as truncated ( ≤ 2 HexNAc; < 3 Hex), paucimannose (2 HexNAc, 3 Hex), high mannose (2 HexNAc; > 3 Hex), hybrid (3 HexNAc) or complex (> 3 HexNAc). Reported peptide spectral matches (PSMs) were pooled across the four parallel protease digestions, from which the fractions of unglycosylated, high-mannose, hybrid, and complex glycans were calculated and averaged over the six replicates. The Python script used for the analysis is provided in the PRIDE deposition.

## Data availability

The atomic models and cryo-EM density maps generated in this study have been deposited to the Protein Data Bank (PDB) (https://www.rcsb.org) and the Electron Microscopy Databank (EMDB) (https://www.ebi.ac.uk/emdb/). The PDB accession numbers are 9CXT, 9CXU, 9D2M, 9D1U, and 9D0Y and the EMDB accession numbers EMD-45997, 45998, 46500 (https://www.ebi.ac.uk/emdb/EMD-45500), 46477, and 46466. The raw LC-MS/MS files and analyses have been deposited to the ProteomeXchange Consortium via the PRIDE partner repository with the dataset identifier PXD051492. Source data are provided in this paper.

## Code availability

All custom scripts have been made available at https://github.com/chazbot72/anchor.

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

## Acknowledgements

We thank B. Anderson and H.L. Turner for cryo-EM data collection support and C. Bowman and J.C. Ducom for computational support. This work was supported, in part, by a Bill and Melinda Gates Foundation grant INV-004923. A.T.d.l.P is a recipient of NWO Rubicon (#45219118). R.d.P.F.R. is supported, in part, by the Pew Charitable Trust and Chan Zuckerberg initiative Foundation (#00036911) and by the Fundação de Amparo à Pesquisa do Estado de São Paulo - FAPESP (#2019/20772-4). R.P.d.V. is a recipient of an ERC Starting Grant from the European Commission (#802780), and a 2023 Research Grant from the Mizutani Foundation for Glycoscience. J.S. and W.P. are funded by the Dutch Research Council NWO Gravitation 2013 BOO, Institute for Chemical Immunology (ICI; 024.002.009). Molecular graphics images were produced using the Chimera package from the Computer Graphics Laboratory, University of California, San Francisco (supported by NIH P41 RR-01081).

## Author contributions

R.d.P.F.R. and A.T.d.l.P. performed sequence analysis, protein purification, BLI, and NS-EM experimental procedures. Also, they did cryo-EM sample preparation, data collection and processing. E.S., J.K., Z.T.B., and S.C performed cryo-EM processing and computational analysis. C.M.P., W.P., and J.S. performed glycoproteomics-based LC-MS/MS experiments and analysis. S.H. provided influenza sera samples. I.T. and D.D.J. performed ELISA and protein expression. S.O., T.G.A., and J.A.F. performed sera processing and EMPEM data analysis. C.A.B. performed computational analysis. R.P.d.V. provided crucial materials; R.d.P.F.R., A.T.d.l.P., S.C., Z.T.B., R.P.d.V., and A.B.W. wrote the paper. All authors contributed to the manuscript text by assisting in writing or providing feedback. A.T.d.l.P., R.E.M., R.P.d.v., Z.T.B., and A.B.W. supervised the research.

## Competing interests

The authors declare no competing interests. Data, materials, and protocols are available upon request to ensure transparency and reproducibility. Images have not been manipulated beyond necessary adjustments. This is original research, not previously published or submitted elsewhere, and any post-publication errors will be promptly addressed with the journal. The work has undergone peer review, and all contributors not meeting authorship criteria are acknowledged.
