## [Transparent Peer Review file · Nature Communications]

Structural and immunological characterization of the H3 influenza hemagglutinin during antigenic drift

Corresponding Author: Dr Andrew Ward

Version 0:

Reviewer comments:

Reviewer #1

(Remarks to the Author)

In this manuscript the authors describe structural and functional perturbations that differential N-linked glycosylation engenders on the influenza hemagglutinin (HA) protein.

Particularly, the authors show that (1) influenza HAs have been accumulation the number of glycosylations from 1968 when H3N2 viruses began to circulate within the human host and today (2) that increased glycosylation of the HA head domain reduces the binding on head-directed bnAbs and shifts polyclonal antibody responses and (3) that enzymatic removal of the glycans results in a structural destabilization of the more glycosylated HA trimer.

In general, the manuscript presents sound data and is relatively well-written. However, outstanding questions remain:

Major:

I don't quite agree with the statement about the interpretation of EMPEP and BLI data in lines 279-281. Indeed, simply because EMPEP with Fab does not show RBS-directed Ab classes, it does not mean those mAbs do not exist. EMPEP is performed at low protein concentrations and with Fab, therefore favoring only high-affinity interactions. Lower affinity Fab or IgG responses certainly exist; indeed the HAI data show higher values for the highly glycosylated HA. Therefore, RBS-directed responses should also be higher. The authors should address this experimentally and conceptually.

Minor:

- 1 - The summary statement mentions challenges in identifying conserved epitopes. This is untrue - stem and RBS epitopes are conserved and protective - the challenges posed are how to best elicit them with a suitable vaccine formulation and regimen.
- 2 - The term vestigial esterase and esterase are used interchangeably throughout the manuscript in the main body and in the figures/legends. It should be 'vestigial esterase' throughout.
- 3 - What are 'glycan heavy atoms'? Presumably non-hydrogen atoms?
- 4 - In Figure 3, if the N22 glycan was not detected in mass spec, how was it modeled? i.e. which type of glycan was used for modeling?
- 5 - In Figure 4, what is the N of technical replicates and independent experiments performed for the data in panel A? Is the difference observed for Sing/16 bnAb binding significant? In the same vein, there is a discrepancy about data interpretation in the main text and the discussion about the reactivity of bnAbs to HK/68 versus Sing/16.
- 6 - In Figure 5D, what are the red versus bkue bars? In general, the figure legends lack detail. Likewise, the methods section should be proof-read as there are typos (e.g., 20 mg/ml protein concentration for EMPEP) and visible track-changes in a couple of instances.

Reviewer #2

(Remarks to the Author)

This is an outstanding paper that will be of wide interest to virologists, viral immunologists, and glycobiologists. Just a few suggestions for the authors to consider.

1. Expand on the differences in immunogenicity in ferrets of the three H3N2 viruses. I would mention that this is not necessarily intrinsic to the HA immunogens but might be related to differences in the infections themselves. To establish the

former, it would be necessary to immunize with recombinant HAs.

2. The effect of HA N-linked glycans on antibody binding has been reported to be temperature-sensitive in some cases (<https://doi.org/10.1371/journal.ppat.1004204>). Would be interesting to examine the effects of temperature (at least 37 degrees, higher as well if enthusiastic) on glycan blockading of Ab...easily examined by HI.

Reviewer #3

(Remarks to the Author)

Ward and co-workers analyzed the ~11,000 sequences of influenza HA protein variants over the decades to demonstrate the preferential increase of N-glycosylation sites within the head region of the HA compared to the stem region. They mapped the changes in N-glycosylation onto the structures of two representative variants, HK/68 and Sing/16, to highlight the effects of shielding by these glycans. The changes in high-mannose contents of the N-glycosylation sites are quantified by glycoproteomics analyses (without technical or biological replicates). They further used a previously developed modelling procedure to estimate the shielding effects of the N-glycans, which are more prominent in the head region. The increased glycan shielding effects may be responsible for shifting the antibody recognition from the head region to the stem region, as demonstrated by the EM-based polyclonal epitope mapping (EMPEM). Finally, the authors compared the high-resolution cryo-EM maps derived from HK/68 and Sing/16 with and without EndoH treatments to show that the removal of N-glycans leads to loss of well-defined EM density within the head region, an observation described as destabilization. The thermal dynamic effect of the EndoH treatment was investigated by differential scanning fluorimetry: there were no significant changes in the melting temperature for HK/68, while some differences were observed for Sing/16. Overall, the study confirmed existing literature about the periodic increase of N-glycosylation sites within the HA protein, and the effect of clustering that results in increased high-mannose contents of the crowded N-glycans. The increased protein surface shielding effects were illustrated by computational modelling as well as cryo-EM data processing to highlight the difference maps corresponding to the N-glycans. Nonetheless, most of the data are confirmative with limited new findings. Therefore, this work would be suited for a more specialized journal.

Major comments

1. The domain definitions shown in Figure 1D should be moved to the very beginning of Figure 1 with 1D domain schematics to highlight the N-glycosylation sites as shown in the current Figure 1A/B to aid visual comparison of the pretty complex figures throughout the manuscript.
2. Lines 132-137: It is not easy to link the stacked graph in Figure 1A with the long descriptions from line 102 to line 131 and to draw the conclusion about the less frequent and more stable glycosylation patterns within the esterase and stem regions.
3. Figure 1C: It is unclear how the structural mapping of the same set of the ~11,000 sequences would lead to very different colouring patterns for HK/68 and Sing/16. Sing/16 also exhibits more N-glycans in the stem region than HK/68, i.e., the increased N-glycosylation sites are not limited to the head region. There are also regions coloured in grey/black within the cavities; it is hard to distinguish between the two structures. This is presumably due to the shadows, but in some places, the dark regions coincide with the N-glycan models, and the glycan models are not labelled. It would be good to improve the presentation to highlight the increased number and density of the N-glycans in the HA of Sing/16.
4. Line 203: The authors should elaborate on how the change in high-mannose content observed for N165 in HA is in line with the previous results reported in ref. 24.
5. Lines 245-248: The authors stated that the different glycosylation profiles of HK/68 and Sing/16 did not impact the stem-specific antibody CR8020 binding, but they do have an impact on the binding of the head-specific antibody, C05. When referring to Figure 5B, there are indeed changes in the OD485 readouts at high antibody concentrations of C05, but it is not apparent how much difference there is in the EC50 values. The corresponding EC50 should be presented and compared to demonstrate their differences are statistically significant.
6. The BLI data presented in Figure 4Ci are not good. They are not quantitatively fit a specific binding model (typically a one-site binding model) to deduce the on- and off-rates to determine the binding affinity in dissociation constants. The binding phase (on-rate) of all three traces is not what is expected for a simple one-site binding. More importantly, the colours of the three traces are not defined, making it difficult to relate to the descriptions made in lines 267-271.
7. The EMPEM analysis indeed showed a stem-binding Fab in Figure 4Civ for the sera against Sing/16 (blue volume) but in the same dataset, a head-binding Fab (purple volume) is also observed. The head-binding Fab binds to HA in a very similar pose as the one observed in the sera against NL03 (Figure 4Ciii). It is, therefore, not a clear cut as stated by the authors. More importantly, the EMPEM results themselves are not sufficient to support the conclusion drawn by the authors in lines 278-281: "Further, our EMPEM data along with our BLI analysis indicate that the polyclonal stem and esterase directed antibodies against Sing/16 not only effectively neutralize the virus but also, are potentially more efficient than the head directed antibodies against HK/68."
8. Supplementary Figure S3 is oversimplified. The data collection and the cascades of particle selections should be described in detail.
9. The thermal denaturation data derived from differential scanning fluorimetry in Figure 5F shows very insignificant transitions for Sing/16 without EndoH treatment, raising the concern of whether the sample was intact and properly folded when subject to the thermal denaturation analysis. Additionally, it is not apparent that multiple transitions can be assigned to specific folding events if the system exhibits multiphasic unfolding processes. Even so, the authors highlighted the second transition peak for the EndoH-treated Sing/16 (red curve) to compare with the very small peak of the untreated Sing/16 (green curve), and the transition temperature for the untreated is, in fact, lower than the treated one. This is contrary to the statement made by the authors.
10. Last but not least, can the authors use the modelling procedure presented in Figure 3A to correlate with the difference

maps shown in Figure 5C to demonstrate the reliability of the modelling results?

Minor comments

1. Lines 76-78: Some references about the essential role of mass spectrometry in defining the N-glycosylations at specific PNGS should be included.
2. Ref. 3 does not mention the glycan addition every 5- years as cross-references in line 97, but ref. 10 does.
3. The figures are not made to the publication standards. Many labels are too small (Figure 1A, Figure 2F/G, Figure 5F, Supporting Figure S2, Supplementary Figure S7A/B). The FSC curve of Sing16/HEK293F is not plotted in the same way as the other four. They should be plotted consistently with high resolutions.
4. Supplementary Figure S4 lists the r.m.s.d. values without units (presumably in Å).
5. Lines 184 and 187: A Greek alpha should replace the "a" of Ca-root-mean-square deviation.
6. Line 310: The authors stated that the cryo-EM maps of EndoH-treated HK/68 and Sing/16 are "nearly identical" to their natively glycosylated counterparts with reference to Figure S3. This should be Figure S4 instead.
7. Lines 311-313: The statement about the more uniform spatial distribution is not adequately supported by the data presented in Figure S3. Quantitative angular distributions of the observed particle images should be given to support this statement.

Reviewer #4

(Remarks to the Author)

1. In the Introduction, the authors correctly pointed out that "the presence of a PNGS is not sufficient to guarantee glycan occupancy". Therefore PNGS and glycans are not interchangeable terms. Yet, in the first section of the Results, prior to the second section on "Defining the structure and glycosylation profile of Sing/16", the word "glycan" was used instead of PNGS in several places. For example,

Ln 96: glycan addition in H3 HAs is estimated to occur every 5 to 7 years - it is unclear if the authors refer only to PNGS, or that the actual additional glycan occupancy has been experimentally verified.

Ln 99: "We analyzed N-glycosidic linkages to the Asn residue of the glycosylation motif Asn-X-Ser/Thr accumulated on HA based on the analysis of ~11,000 sequences of naturally occurring H3 strains using the code previously described by Wu et al." - It is unclear if the authors refer only to PNGS or have actually analyzed the glycans.

Ln 108: "Interestingly, these glycans all appear on esterase or stem regions of HA except for N165, which is in the globular head at the interface between protomers (Figure 1B). Other glycans that also appear on esterase or stem regions on HA such as N45, N63, N81 and N276, appear transiently (Figure 1A-B)." - it would seem that the authors actually meant only the PNGS and NOT glycans.

Ln 120: "By analyzing the glycans present on A/Hong Kong/1/1968 (HK/68) Again, here the authors described and refer to Fig 1 when it appears to be referring to PNGS only.

2. Ln 197: "We lack coverage at N22 and N81 on HK/68, and N45 and N63 on Sing/16, either due to low peptide detection or poor fragmentation." - the authors could, in principle, look at the MS dataset and verify if any of the non-glycosylated peptides were detected versus any potential glycosylated versions. If it is due to poor MS2 fragmentation, the authors could verify if putative glycopeptide peaks were present at MS1 level. In fact, the authors did not address the degree of site occupancy. Analytically, this is not trivial. The authors should, however, be more precise in their choice of words. What does it mean by "low peptide detection" or "poor fragmentation"? In contrast, the authors appear to have detected non-glycosylated N122 peptide from the Sing/16 sample but neither the glycosylated nor the unglycosylated N122 peptide was detected in the HK/68 sample. Is it not detected or low peptide detection, or detected but poor fragmentation?

3. No Suppl data on the glycopeptide identification by Byonic search results was provided and hence the quality cannot be assessed. The authors stated that "The raw LC-MS/MS files and analyses have been deposited to the ProteomeXchange Consortium via the PRIDE partner repository with the dataset identifier PXD051492" A quick look up using the identifier cannot find any matching dataset.

The statement in the Methods section: All reported glycopeptides in the Byonic result files were manually inspected for quality of fragment assignments (with scores ≥ 200). This is not a very convincing criterion although commonly adopted in the field. For current purposes (Fig 2F, G), the number of mismatches or misassignments would perhaps have no significant consequence on the big picture. It should be noted that to classify glycosyl composition with HexNAc=3 as hybrid type can be misleading. HexNAc3 can also be contributed by complex type biantennary glycans with diLacNAc or lacdiNAc on one antenna, or a triantennary glycan.

4. It will be good to do similar LC-MS/MS analysis of endoH-treated sample and see if those sites classified as carrying the highest proportion of high mannose structures are indeed those most affected.

Version 1:

Reviewer comments:

Reviewer #1

(Remarks to the Author)
No further comments.

Reviewer #3

(Remarks to the Author)
The authors have sufficiently addressed all my comments and I would recommend the acceptance of the manuscript in its current form.

Reviewer #4

(Remarks to the Author)
I have no further questions on the issues I raised. I do acknowledge that I made a mistake in writing that "HexNAc3 can also be contributed by complex type biantennary glycans with diLacNAc or lacdiNAc on one antenna, or a triantennary glycan." What I had in mind and should actually say is that HexNAc^{>3}, specifically HexNAc⁼⁴, may still be a hybrid type in situation when the extra HexNAc is contributed by a diLacNAc or lacdiNAc on the single extended antenna, or a bisecting GlcNAc. In any case, it is true that this glycan classification is widely adopted although it may misclassify some of the hybrid type structures as complex type.

Response to Reviewers for Manuscript NCOMMS-24-73049

(Original reviewers' comments in black, our response in green)

Reviewer #1 (Remarks to the Author):

In this manuscript the authors describe structural and functional perturbations that differential N-linked glycosylation engenders on the influenza hemagglutinin (HA) protein. Particularly, the authors show that (1) influenza HAs have been accumulation the number of glycosylations from 1968 when H3N2 viruses began to circulate within the human host and today (2) that increased glycosylation of the HA head domain reduces the binding on head-directed bnAbs and shifts polyclonal antibody responses and (3) that enzymatic removal of the glycans results in a structural destabilization of the more glycosylated HA trimer.

In general, the manuscript presents sound data and is relatively well-written.

We sincerely appreciate the positive feedback from the reviewer on the manuscript.

However, outstanding questions remain:

Major:

I don't quite agree with the statement about the interpretation of EMPEP and BLI data in lines 279-281. Indeed, simply because EMPEM with Fab does not show RBS-directed Ab classes, it does not mean those mAbs do not exist. EMPEM is performed at low protein concentrations and with Fab, therefore favoring only high-affinity interactions. Lower affinity Fab or IgG responses certainly exist; indeed, the HAI data show higher values for the highly glycosylated HA. Therefore, RBS-directed responses should also be higher. The authors should address this experimentally and conceptually.

We thank the reviewer for the helpful comments. In response, we have added competition ELISA data using both Fabs and polyclonal IgG, to further clarify this point. As described in line 269-279 and Figure 4B-D, competition for the head domain is increased in HK/68 and

NL03 compared to Sing/16, highlighting the impact of glycosylation on antibody binding. As suggested by the reviewer, we attempted to perform EMPEM using polyclonal IgG. However, we observed substantial aggregation and crosslinking between IgGs, which prevented successful image processing. Therefore, we optimized our EMPEM protocol to improve Fab-antigen complex recovery. Specifically, after complexing the antigen with Fabs, samples were not passed through a SEC column and instead we removed unbound Fabs using 100 kDa Amicon filters and centrifuging at slow speed (Methods line 521-526). This adjustment enabled us to confirm Fab binding to the head of HK/68 and to detect head-binding Fabs to NL03 that were not previously observed (Figure 4B-C). However, we still did not detect Fab binding to the head of Sing/16. Additionally, we performed a competition ELISA using a panel of bNAbs against the RBS (S139/1, K03.28, TJ5-5, C05 and F045-92) and against the stem (CR9114) and using polyclonal IgG and Fabs isolated from reference ferret sera (Figure 4A). Similarly to the EMPEM data, while we observed competition of RBS antibodies and HK/68 and NL/03 sera, we only observed competition of stem antibodies in Sing/16 sera. Thus, while Fabs targeting the head domain may be present in the Sing/16 response, our EMPEM and competition ELISA data suggest they are of low abundance (Figure 4).

Minor:

1 - The summary statement mentions challenges in identifying conserved epitopes. This is untrue - stem and RBS epitopes are conserved and protective - the challenges posed are how to best elicit them with a suitable vaccine formulation and regimen.

We thank the reviewer for pointing this out, the revised manuscript includes a sentence in the summary section with the correct statement (line 33-34).

2 - The term vestigial esterase and esterase are used interchangeably throughout the manuscript in the main body and in the figures/legends. It should be 'vestigial esterase' throughout.

We thank the reviewer for this suggestion. The correct name is vestigial esterase and this has now been changed through the text and figures.

3 - What are 'glycan heavy atoms'? Presumably non-hydrogen atoms?

We thank the reviewer for raising this question. Indeed, in modeling, 'heavy atoms' refers to all non-hydrogen atoms. This has now been clarified (line 223 and lines 662-663).

4 - In Figure 3, if the N22 glycan was not detected in mass spec, how was it modeled? i.e. which type of glycan was used for modeling?

The reviewer raises an important point on how the modelling was done without mass spectrometry data. In that case, the most common glycan type for these structures (FA2) was used (Line 724-743).

5 - In Figure 4, what is the N of technical replicates and independent experiments performed for the data in panel A? Is the difference observed for Sing/16 bnAb binding significant? In the same vein, there is a discrepancy about data interpretation in the main text and the discussion about the reactivity of bnAbs to HK/68 versus Sing/16.

We thank the reviewer for pointing this out. Experiments were done in duplicate and this is now stated in the legend of Figure 4 and S5 and S6. To test the significance of antibody binding between recombinantly expressed hemagglutinins in 293T, GnTI- and endo-H treated in a panel of five head antibodies and one stem antibody, we did an ELISA and found that there are minimal, insignificant differences regarding the affinity of the head and stem antibodies between differently glycosylated HAs. We have added a panel in Figure 4 (panel A) and Supplementary Figure 5 with the broader panel of head and stem antibodies and wrote the results and discussion accordingly (lines 243-249). Finally, we have revised the text to clarify the interpretation of bnAb reactivity to HK/68 versus Sing/16 and have also updated the corresponding figure (Figure 4A and S5) to ensure consistency with the revised interpretation (Page 11-14)

6 - In Figure 5D, what are the red versus blue bars? In general, the figure legends lack detail. Likewise, the methods section should be proof-read as there are typos (e.g., 20 mg/ml protein concentration for EMPEM) and visible track-changes in a couple of instances.

We thank the reviewer for pointing this out. We have added more detail in Figure 5 panel D and in the figure legend. Additionally, we incorporated a supplementary figure to have more context on panel C (Supplementary Figure 8). The supplementary figure shows the simulated cryo-EM volumes from atomistic ensembles (Figure S8). Specifically, it shows synthetic cryo-EM density maps generated from the 1000 atomistic models of fully glycosylated HK/68 (blue) and Sing/16 (green) HAs are shown at high and low contour levels from the top and side (Figure S8 and Figure 3). Also, we have proof-read the methods sections and polished it accordingly.

Reviewer #2 (Remarks to the Author):

This is an outstanding paper that will be of wide interest to virologists, viral immunologists, and glycobiochemists.

We sincerely appreciate the positive feedback and enthusiasm from the reviewer on the manuscript.

Just a few suggestions for the authors to consider.

1. Expand on the differences in immunogenicity in ferrets of the three H3N2 viruses. I would mention that this is not necessarily intrinsic to the HA immunogens but might be related to differences in the infections themselves. To establish the former, it would be necessary to immunize with recombinant HAs.

The reviewer raises a valid point, and we have changed the text accordingly (Line 257-271)

We have further addressed this point in the discussion (Lines 364–371).

2. The effect of HA N-linked glycans on antibody binding has been reported to be temperature-sensitive in some cases (<https://doi.org/10.1371/journal.ppat.1004204>). Would be interesting to examine the effects of temperature (at least 37 degrees, higher as well if enthusiastic) on glycan blockading of Ab...easily examined by HI.

We agree with the reviewer that it would be interesting to examine the effects of temperature on glycan blockading of antibodies, however, we believe it is beyond the scope of this paper.

Reviewer #3 (Remarks to the Author):

Ward and co-workers analyzed the ~11,000 sequences of influenza HA protein variants over the decades to demonstrate the preferential increase of N-glycosylation sites within the head region of the HA compared to the stem region. They mapped the changes in N-glycosylation onto the structures of two representative variants, HK/68 and Sing/16, to highlight the effects of shielding by these glycans. The changes in high-mannose contents of the N-glycosylation sites are quantified by glycoproteomics analyses (without technical or biological replicates).

The original glycoproteomics analyses are based on the combined findings of four parallel protease digestions. While it is true that each was analyzed by a single LC-MS/MS run, the redundant coverage across the different protease digestions does make for a more robust estimate of the relative glycan profiles across the N-linked glycosylation sites of HA. That being said, we appreciate the reviewer's concerns and opted to redo our glycoproteomics analyses with additional replicates. The revised manuscript now reports glycoproteomics findings of four parallel protease digestions performed in duplicate, each in technical triplicate, for a total of six replicate analyses per sample. These new analyses replace the originally reported glycoproteomics experiments as the original MS instrument used for the first experiments was recently decommissioned. The newly reported experiments are performed on an improved OrbiTrap TriBid mass spectrometer (Orbitrap Eclipse). The new findings are fully in line with the original experiments; if anything, we find a more pronounced effect of increased underprocessed high-mannose glycans in the Sing/16 sample (Figure 2F-G).

They further used a previously developed modelling procedure to estimate the shielding effects of the N-glycans, which are more prominent in the head region. The increased glycan shielding effects may be responsible for shifting the antibody recognition from the head region to the stem region, as demonstrated by the EM-based polyclonal epitope mapping

(EMPEM). Finally, the authors compared the high-resolution cryo-EM maps derived from HK/68 and SIng/16 with and without EndoH treatments to show that the removal of N-glycans leads to loss of well-defined EM density within the head region, an observation described as destabilization. The thermal dynamic effect of the EndoH treatment was investigated by differential scanning fluorimetry: there were no significant changes in the melting temperature for HK/68, while some differences were observed for Sing/16. Overall, the study confirmed existing literature about the periodic increase of N-glycosylation sites within the HA protein, and the effect of clustering that results in increased high-mannose contents of the crowded N-glycans. The increased protein surface shielding effects were illustrated by computational modelling as well as cryo-EM data processing to highlight the difference maps corresponding to the N-glycans. Nonetheless, most of the data are confirmative with limited new findings. Therefore, this work would be suited for a more specialized journal.

We thank the reviewer for the thoughtful feedback. While our results confirm general trends previously reported in the literature, such as the progressive addition of N-glycosylation sites and the clustering of high-mannose glycans, our study goes further by integrating cryo-EM, polyclonal epitope mapping, computational modeling, glycoproteomics, and immunoassays to examine how these changes influence antibody binding and HA structure. This approach goes beyond previous reports by linking glycosylation patterns to functional antibody binding and competition in a structurally resolved context. We hope the reviewer will agree that this mechanistic perspective advances our understanding of HA evolution and immunogenicity in a way that is relevant to virology, structural biology and influenza vaccine design.

Major comments

1. The domain definitions shown in Figure 1D should be moved to the very beginning of Figure 1 with 1D domain schematics to highlight the N-glycosylation sites as shown in the current Figure 1A/B to aid visual comparison of the pretty complex figures throughout the manuscript.

We agree with the reviewer and have rearranged Figure 1 accordingly. The domain definitions originally shown in Figure 1D have been moved to the beginning of the figure to aid in visual comparison and improve clarity in the presentation of N-glycosylation sites across the HA structure.

2. Lines 132-137: It is not easy to link the stacked graph in Figure 1A with the long descriptions from line 102 to line 131 and to draw the conclusion about the less frequent and more stable glycosylation patterns within the esterase and stem regions.

We appreciate the reviewers' comments and the paragraph was rewritten and restructured for better clarity (Lines 107-130).

3. Figure 1C: It is unclear how the structural mapping of the same set of the ~11,000 sequences would lead to very different colouring patterns for HK/68 and Sing/16. Sing/16 also exhibits more N-glycans in the stem region than HK/68, i.e., the increased N-glycosylation sites are not limited to the head region. There are also regions coloured in grey/black within the cavities; it is hard to distinguish between the two structures. This is presumably due to the shadows, but in some places, the dark regions coincide with the N-glycan models, and the glycan models are not labelled. It would be good to improve the presentation to highlight the increased number and density of the N-glycans in the HA of Sing/16.

We thank the reviewer for raising this point. We agree that the conservation patterns between HK/68 and Sing/16 are very similar because even if the amino acid sequence is different the conservation per residue will be similar. For clarity and because glycosylation differs between strains, we show both HK68 and Sing16 and we have changed the shadowing to make sure all the glycans are visible and labelled (Figure 1D).

4. Line 203: The authors should elaborate on how the change in high-mannose content observed for N165 in HA is in line with the previous results reported in ref. 24.

We thank the reviewer for the comment and we have addressed the issue in the text (line 202-207).

5. Lines 245-248: The authors stated that the different glycosylation profiles of HK/68 and Sing/16 did not impact the stem-specific antibody CR8020 binding, but they do have an impact on the binding of the head-specific antibody, C05. When referring to Figure 5B, there are indeed changes in the OD485 readouts at high antibody concentrations of C05, but it is not apparent how much difference there is in the EC50 values. The corresponding EC50 should be presented and compared to demonstrate their differences are statistically significant.

We thank the reviewer for raising this point. To test the significance of antibody binding between recombinantly expressed hemagglutinins in 293T, GnTI- and endo-H treated in a panel of five head antibodies and one stem antibody, we did multiple ELISAs using a panel of RBS binding antibodies as well as one stem antibody. We calculated the EC50s and the AUC and observed minimal differences, not significant, regarding the affinity of the head and stem antibodies between differently glycosylated HAs. Thus, RBS antibodies bind similarly between strains and differently glycosylated HAs. We have added a panel in Figure 4 (Figure 4A) with the panel of head and stem antibodies and wrote the results and discussion accordingly (243-249).

6. The BLI data presented in Figure 4Ci are not good. They are not quantitatively fit a specific binding model (typically a one-site binding model) to deduce the on- and off-rates to determine the binding affinity in dissociation constants. The binding phase (on-rate) of all three traces is not what is expected for a simple one-site binding. More importantly, the colours of the three traces are not defined, making it difficult to relate to the descriptions made in lines 267-271.

We thank the reviewer for pointing this out. While we agree that when using monoclonal antibodies, we can calculate dissociation constants because each antibody binds one epitope of the antigen at a specific affinity and dissociation constant, here, we tested a polyclonal antibody sample against one antigen. A polyclonal antibody sample contains

antibodies with several specificities, specifically, antibodies that bind different epitopes. Because of that, we cannot calculate the association and dissociation constants. BLI was performed using polyclonal antibody samples as a control and to assess binding of the antibodies to the match antigens.

7. The EMPEM analysis indeed showed a stem-binding Fab in Figure 4Civ for the sera against Sing/16 (blue volume) but in the same dataset, a head-binding Fab (purple volume) is also observed. The head-binding Fab binds to HA in a very similar pose as the one observed in the sera against NL03 (Figure 4Ciii). It is, therefore, not a clear cut as stated by the authors. More importantly, the EMPEM results themselves are not sufficient to support the conclusion drawn by the authors in lines 278-281: “Further, our EMPEM data along with our BLI analysis indicate that the polyclonal stem and esterase directed antibodies against Sing/16 not only effectively neutralize the virus but also, are potentially more efficient than the head directed antibodies against HK/68.” The purple (now pink) volume depicts the vestigial esterase epitope (Figure 4B-D). This region is within the head domain but antibodies binding to that region are a distinct class of antibodies compared to the stem epitope, as has been previously defined in the literature (Lines 96-97) For clarity, we have included a schematic representation of the epitopes targeted and the name of each epitope in a panel of Figure S6C.

We have changed this statement and added data from competition ELISA (see reviewer’s 1 answer to question 1) for clarity (Line 272-279).

8. Supplementary Figure S3 is oversimplified. The data collection and the cascades of particle selections should be described in detail.

We thank the reviewer for pointing this out. We have updated Figure S3 and expanded the description of data collection and processing to provide additional detail.

9. The thermal denaturation data derived from differential scanning fluorimetry in Figure 5F shows very insignificant transitions for Sing/16 without EndoH treatment, raising the concern of whether the sample was intact and properly folded when subject to the thermal

denaturation analysis. Additionally, it is not apparent that multiple transitions can be assigned to specific folding events if the system exhibits multiphasic unfolding processes. Even so, the authors highlighted the second transition peak for the EndoH-treated Sing/16 (red curve) to compare with the very small peak of the untreated Sing/16 (green curve), and the transition temperature for the untreated is, in fact, lower than the treated one. This is contrary to the statement made by the authors.

We thank the reviewer for raising these points. We have assessed the conformation of the HA trimers before nano-DSF by negative stain EM and have added it to the supplementary data (Figure S6B). Additionally, we have repeated the experiment with the same amount of protein and getting similar size peaks in nano-DSF and have adjusted the statement with the new results, similar to the ones we had before (Figure 5F).

10. Last but not least, can the authors use the modelling procedure presented in Figure 3A to correlate with the difference maps shown in Figure 5C to demonstrate the reliability of the modelling results?

We thank the reviewer for raising this point. We incorporated a supplementary figure to have more context on panel C. The supplementary figure shows the simulated cryo-EM volumes from atomistic ensembles, specifically, it shows synthetic cryo-EM density maps generated from the 1000 atomistic models (Figure 3) of fully glycosylated HK/68 (blue) and SING/16 (green) HAs shown at high and low contour levels from the top and side (Figure S8) (Line 315-319)

Minor comments

1. Lines 76-78: Some references about the essential role of mass spectrometry in defining the N-glycosylations at specific PNGS should be included.

We thank the reviewer for the helpful comment. We have added two additional references to highlight the essential role of mass spectrometry in defining N-glycosylation at specific PNGSs (Page 3; Line 76; Ref 14,15).

2. Ref. 3 does not mention the glycan addition every 5- years as cross-references in line 97, but ref. 10 does. We thank the reviewer for the helpful comment and have fixed the issue.

3. The figures are not made to the publication standards. Many labels are too small (Figure 1A, Figure 2F/G, Figure 5F, Supporting Figure S2, Supplementary Figure S7A/B). The FSC curve of Sing16/HEK293F is not plotted in the same way as the other four. They should be plotted consistently with high resolutions.

We thank the reviewer for the helpful comment and have corrected the references. Regarding the figures, we agree with the reviewer that several elements were not initially up to publication standards. We have revised Figures 1A, 2F/G, 4, 5F, Supplementary Figure 2, which we added the excel graph as supplementary data 1, and S9A/B (old S7A/B) to improve label size and clarity.

We also acknowledge the inconsistency in the FSC plot for Sing/16 produced in HEK293T cells. This discrepancy arises because this map was processed using RELION 4.0, while the other reconstructions were processed in cryoSPARC. We have added this information in the figure legend for clarity (Figure S2C).

4. Supplementary Figure S4 lists the r.m.s.d. values without units (presumably in Å). We thank the reviewer for pointing this out. We have added the units (Å) to the figure and the figure legend.

5. Lines 184 and 187: A Greek alpha should replace the “a” of Ca-root-mean-square deviation. We have replaced the a in the text (Line 184 and 653).

6. Line 310: The authors stated that the cryo-EM maps of EndoH-treated HK/68 and Sing/16 are “nearly identical” to their natively glycosylated counterparts with reference to Figure S3. This should be Figure S4 instead. We thank the reviewer for pointing this out. We have changed that in the text. It is now S3 because we rearranged supplementary figures.

7. Lines 311-313: The statement about the more uniform spatial distribution is not

adequately supported by the data presented in Figure S3. Quantitative angular distributions of the observed particle images should be given to support this statement.

We thank the reviewer for this comment. To address this, we have added view distribution plots from CryoSPARC to Figure S2 (panel D), which quantitatively illustrate the spatial distribution of particle orientations.

Reviewer#4 (Remarks to the Author):

1. In the Introduction, the authors correctly pointed out that “the presence of a PNGS is not sufficient to guarantee glycan occupancy”. Therefore PNGS and glycans are not interchangeable terms. Yet, in the first section of the Results, prior to the second section on “Defining the structure and glycosylation profile of Sing/16”, the word “glycan” was used instead of PNGS in several places. For example,

Ln 96: glycan addition in H3 HAs is estimated to occur every 5 to 7 years - it is unclear if the authors refer only to PNGS, or that the actual additional glycan occupancy has been experimentally verified. We thank the reviewer for this important note. The statement is in reference to published work see ref Altman et al., mBio 2019 “Human Influenza A Virus Hemagglutinin Glycan Evolution Follows a Temporal Pattern to a Glycan Limit”, which combined sequence analysis with biochemical methods to estimate the periodic acquisition of glycan on HA (Line 101).

Ln 99: “We analyzed N-glycosidic linkages to the Asn residue of the glycosylation motif Asn-X-Ser/Thr accumulated on HA based on the analysis of ~11,000 sequences of naturally occurring H3 strains using the code previously described by Wu et al.” - It is unclear if the authors refer only to PNGS or have actually analyzed the glycans. We thank the reviewer for the comment. We have revised the text for clarity to specify that, in Figure 1, our analysis was limited to the presence of predicted PNGSs.

Ln 108: “Interestingly, these glycans all appear on esterase or stem regions of HA except for N165, which is in the globular head at the interface between protomers (Figure 1B). Other

glycans that also appear on esterase or stem regions on HA such as N45, N63, N81 and N276, appear transiently (Figure 1A-B).” - it would seem that the authors actually meant only the PNGS and NOT glycans. We thank the reviewer for the comment and have restructured the text to improve clarity (lines 107-130).

Ln 120: “By analyzing the glycans present on A/Hong Kong/1/1968 (HK/68) Again, here the authors described and refer to Fig 1 when it appears to be referring to PNGS only.

We thank the reviewer for the comment and have restructured the text to improve clarity.

2. Ln 197: “We lack coverage at N22 and N81 on HK/68, and N45 and N63 on Sing/16, either due to low peptide detection or poor fragmentation.” - the authors could, in principle, look at the MS dataset and verify if any of the non-glycosylated peptides were detected versus any potential glycosylated versions. If it is due to poor MS2 fragmentation, the authors could verify if putative glycopeptide peaks were present at MS1 level. In fact, the authors did not address the degree of site occupancy. Analytically, this is not trivial. The authors should, however, be more precise in their choice of words. What does it mean by “low peptide detection” or “poor fragmentation”? In contrast, the authors appear to have detected non-glycosylated N122 peptide from the Sing/16 sample but neither the glycosylated nor the unglycosylated N122 peptide was detected in the HK/68 sample. Is it not detected or low peptide detection, or detected but poor fragmentation?

The database search included identification of the unmodified asparagine. When we identified it, it is reported as such. The missing sites should not be interpreted as lacking a glycan, but rather as blind spots in the experiment. Occupancy is in fact addressed in the original figure like for N122 in Sing/16 sample. For the other sites no unoccupied asparagines were identified; to the best of our knowledge they are completely occupied up to the detection limit of our method. We are hesitant to infer anything from MS1 levels alone because it yields too many false positives without opportunities to validate IDs based on fragmentation patterns. Please note that the glycoproteomics analyses have been redone on a new instrument platform with additional replicate analyses. This has improved our

coverage to now also include N45 in Sing/16 and we recovered additional unglycosylated asparagines at sites beyond N122 in both HK/68 and Sing/16 samples.

3. No Suppl data on the glycopeptide identification by Byonic search results was provided and hence the quality cannot be assessed. The authors stated that “The raw LC-MS/MS files and analyses have been deposited to the ProteomeXchange Consortium via the PRIDE partner repository with the dataset identifier PXD051492” A quick look up using the identifier cannot find any matching dataset.

We regret that the PRIDE login details have not been made available earlier. The data can be accessed under identifier PXD051492 using the following login details:

Reviewer account details:

Username: reviewer_pxd051492@ebi.ac.uk

Password: 4o1VxgX7

The statement in the Methods section: All reported glycopeptides in the Byonic result files were manually inspected for quality of fragment assignments (with scores ≥ 200). This is not a very convincing criterion although commonly adopted in the field. For current purposes (Fig 2F, G), the number of mismatches or misassignments would perhaps have no significant consequence on the big picture. It should be noted that to classify glycosyl composition with HexNAc=3 as hybrid type can be misleading. HexNAc3 can also be contributed by complex type biantennary glycans with diLacNAc or lacdiNAc on one antenna, or a triantennary glycan.

We thank the reviewer for this insightful comment. As stated by the reviewer the score cutoff is widely used. Moreover, it is a lower limit. The scores range from 200 to 1717, with a mean/median of 549/483 across the datasets for both HK/68 and Sing/16

There appears to be a misunderstanding about the hybrid classification. The counts of HexNAc residues include the core pentasaccharide, so we start counting at 2. Hybrids have +1 on one arm (exactly 3), complex +1 for every antenna bringing the total to ≥ 4 . The only risk we have in this classification is if the processed arm of the hybrid glycans contain

diLacNAc or LacdiNAc to bring its total to 4 or higher, very rare and the converse of what the reviewer suggests. This glycan classification is widely adopted in the glycoproteomics field.

4. It will be good to do similar LC-MS/MS analysis of endoH-treated sample and see if those sites classified as carrying the highest proportion of high mannose structures are indeed those most affected.

We thank the reviewer for the suggestion. While LC-MS/MS analysis of Endo H-treated samples could in principle provide further support for the high-mannose classification at specific sites, we believe this additional validation is not essential for the current study. The assignment of high-mannose glycoforms was based on well-established MS-based workflows and stringent manual curation, consistent with standard practices in the glycoproteomics field.